# Sharper Convergence Guarantees for Asynchronous SGD for Distributed and Federated Learning

**Anastasia Koloskova**
EPFL
anastasia.koloskova@epfl.ch

**Sebastian U. Stich**
CISPA*
stich@cispa.de

**Martin Jaggi**
EPFL
martin.jaggi@epfl.ch

## Abstract

We study the asynchronous stochastic gradient descent algorithm for distributed training over $n$ workers which have varying computation and communication frequency over time. In this algorithm, workers compute stochastic gradients in parallel at their own pace and return those to the server without any synchronization. Existing convergence rates for this algorithm for non-convex smooth objectives depend on the maximum gradient delay $\tau_{\max}$ and show that an $\varepsilon$-stationary point is reached after $\mathcal{O}\big(\sigma^2\varepsilon^{-2} + \tau_{\max}\varepsilon^{-1}\big)$ iterations, where $\sigma$ denotes the variance of stochastic gradients.

In this work we obtain (i) a tighter convergence rate of $\mathcal{O}\big(\sigma^2\varepsilon^{-2} + \sqrt{\tau_{\max}\tau_{avg}}\varepsilon^{-1}\big)$ *without any change in the algorithm*, where $\tau_{avg}$ is the average delay, which can be significantly smaller than $\tau_{\max}$. We also provide (ii) a simple delay-adaptive learning rate scheme, under which asynchronous SGD achieves a convergence rate of $\mathcal{O}\big(\sigma^2\varepsilon^{-2} + \tau_{avg}\varepsilon^{-1}\big)$, and does not require any extra hyperparameter tuning nor extra communications. Our result allows to show *for the first time* that asynchronous SGD is *always faster* than mini-batch SGD. In addition, (iii) we consider the case of heterogeneous functions motivated by federated learning applications and improve the convergence rate by proving a weaker dependence on the maximum delay compared to prior works. In particular, we show that the heterogeneity term in convergence rate is only affected by the average delay within each worker.

## 1   Introduction

The stochastic gradient descent (SGD) algorithm [43, 13] and its variants (momentum SGD, Adam, etc.) form the foundation of modern machine learning and frequently achieve state of the art results. With recent growth in the size of models and available training data, parallel and distributed versions of SGD are becoming increasingly important [57, 17, 16]. Without those, modern state-of-the art language models [44], generative models [40, 41], and many others [50] would not be possible. In the distributed setting, also known as data-parallel training, optimization is distributed over many compute devices working in parallel (e.g. cores, or GPUs on a cluster) in order to speed up training. Every worker computes gradients on a subset of the training data, and the resulting gradients are aggregated (averaged) on a server.

The same type of SGD variants also form the core algorithms for federated learning applications [34, 24] where the training process is naturally distributed over many user devices, or clients, that keep their local data private, and only transfer (e.g. encrypted or differentially private) gradients to the server.

A rich literature exists on the convergence theory of above mentioned parallel SGD methods, see e.g. [17, 13] and references therein. Plain parallel SGD still faces many challenges in practice, motivat-

---

*CISPA Helmholtz Center for Information Security

36th Conference on Neural Information Processing Systems (NeurIPS 2022).

ing research on various approaches to improve efficiency of distributed learning and mini-batch SGD. This includes for example communication compression techniques [2, 4, 48, 49], decentralized communication [29, 7, 36, 26] or performing several local SGD steps on workers before communicating with the server [30, 32, 34, 47].

These approaches use synchronous communication, where workers in each round are required to wait for the slowest one, before being able to start the next round of computations. In the presence of such straggler nodes or nodes that have different computation speeds, other workers face significant idle times. *Asynchronous* variants of SGD are aimed to solve such inefficiencies and use available workers more effectively. In asynchronous SGD, each worker starts the next computation immediately after finishing computing its own gradient, without waiting for any other workers. This is especially important in the presence of straggler nodes. Asynchronous algorithms were studied both in distributed and federated learning settings [42, 31, 27, 46, 37]. In this paper we focus on such challenging asynchronous variants of SGD and provide an improved theoretical analysis of convergence compared to prior works.

Most existing work has studied the convergence behavior of asynchronous SGD for the setting of homogeneous distributed training data, where worker's objectives are i.i.d.. This assumption however is only realistic e.g. in shared-memory implementations where all processes can access the same data [42]. Under this assumption, it can be proven that asynchronous SGD finds an $\varepsilon$-approximate stationary point (squared gradient norm bounded by $\varepsilon$) in $\mathcal{O}\big(\frac{\sigma^2}{\varepsilon^2} + \frac{\tau_{\max}}{\varepsilon}\big)$ iterations [48], for smooth non-convex objective functions. This complexity bound depends on the maximum delay of the gradients $\tau_{\max}$ and the gradient variance $\sigma > 0$. Unfortunately, the maximal delay is a very pessimistic metric, not well reflecting the true behavior in practice. For instance, if a worker struggles just once, the maximum delay is large, while we would still expect reasonable overall convergence.

Two recent works [15, 9] tackle this issue by proposing two new delay-adaptive algorithms that achieve a convergence rate that depends only on the average delay of the applied gradients, with Aviv et al. [9] considering only the convex optimization and Cohen et al. [15] providing a rate of $\mathcal{O}\big(\frac{\sigma^2}{\varepsilon^2} + \frac{\tau_{avg}}{\varepsilon}\big)$ for smooth non-convex functions. The average delay can be much smaller than the maximal delay, and thus these methods are robust to rare stragglers. However, Cohen et al. [15] requires twice more communications at every step, and an extra hyperparameter to tune. Aviv et al. [9] analyze only convex functions and assume a bound on the variance of the delays, which can frequently degrade with the maximum delay $\tau_{\max}$. Moreover, both works require the assumption that gradients are uniformly bounded.

In the realistic case of heterogeneous objective functions, that is in particular relevant in federated learning applications [24], all the existent convergence rates of asynchronous SGD depend on the maximum delay [37].

**Contributions.**

- For standard asynchronous SGD with constant stepsize, and with non-convex $L$-smooth homogeneous objective functions, we prove the tighter convergence rate of $\mathcal{O}\big(\frac{\sigma^2}{\varepsilon^2} + \frac{\sqrt{\tau_{avg}\tau_{\max}}}{\varepsilon}\big)$ to $\varepsilon$-small error. Under the additional assumption of bounded gradients, we obtain a convergence rate of $\mathcal{O}\big(\frac{\sigma^2}{\varepsilon^2} + \frac{\tau_{avg}G}{\varepsilon^{3/2}} + \frac{\tau_{avg}}{\varepsilon}\big)$ where $G$ is the bound on the norm of gradients. The previously best known rate was $\mathcal{O}\big(\frac{\sigma^2}{\varepsilon^2} + \frac{\tau_{\max}}{\varepsilon}\big)$.

- With homogeneous objective functions, we provide a delay-adaptive stepsize scheme that does not require tuning of any extra hyperparameters, and converges at the rate of $\mathcal{O}\big(\frac{\sigma^2}{\varepsilon^2} + \frac{\tau_{avg}}{\varepsilon}\big)$ for non-convex $L$-smooth functions.

- This result allows us to show that asynchronous SGD is always better than mini-batch SGD regardless of the delays pattern (under assumption that the server can perform operations with zero time).

- We also consider distributed optimization with heterogeneous objectives where the delays can depend on the workers and give the convergence rate of $\mathcal{O}\big(\frac{\sigma^2}{\varepsilon^2} + \frac{\zeta^2}{\varepsilon^2} + \frac{\sqrt{\tau_{avg}\frac{1}{n}\sum_{i=1}^{n}\zeta_i^2\bar{\tau}_{avg}^i}}{\varepsilon^{\frac{3}{2}}} + \frac{\sqrt{\tau_{avg}\tau_{\max}}}{\varepsilon}\big)$, where $\zeta_i$'s measure functions heterogeneity and $\bar{\tau}_i$ is the average delay of worker $i$. This rate improves over the best previously-known results that had worse dependence on the maximum delay $\tau_{\max}$.

## 2 Related Work

**Asynchronous SGD.** The research field of asynchronous optimization can be traced back at least to 1989 [11]. Recent works are heavily focused on its SGD variants, such as Hogwild! SGD [39] which deals with coordinate-wise asynchronity. Nguyen et al. [38] provided a tighter convergence analysis by removing the bounded gradient assumption. Our work does not focus on such a coordinate-wise asynchrony as it relies on sparsity assumption that is not realistic in modern machine learning applications. Mania et al. [31] introduces the perturbed iterate framework which enabled theoretical advances with tighter convergence rates [48, 46]. Leblond et al. [27] focus on asynchronous variance-reduced methods.

Many works [1, 14, 20, 6, 45, 28, 48, 18] focused on asynchronous SGD variants where workers communicate with the server without any synchronization, but these communications are considered to be atomic. All of these works provide convergence guarantees that depend on the maximum delay $\tau_{\max}$ with [6, 48] providing the first tight convergence rates under assumption that the delays are always constant for quadratic and general (convex, strongly convex and non-convex) functions correspondingly. Stich et al. [46] showed a connection of large batches and delays, although still depending only on the maximum delay. Even et al. [19] consider a continuized view of the time (rather than classical per-iteration time) for asynchronous algorithms on a decentralized network. [3] improve the convergence rates of asynchronous SGD to depend on $\sqrt{\tau_{\max}n}$, however assuming bounded gradients. Under bounded gradients our convergence rates completely remove dependence on the maximum delay $\tau_{\max}$.

**Delay-adaptive methods.** The works [56, 55, 45, 51, 33, 18] considered delay-adaptive schemes to mitigate adversarial effect of stragglers, however with convergence rates that still depend on the maximum delay $\tau_{\max}$. Only Cohen et al. [15] in the non-convex, and Aviv et al. [9] in the convex case were able to obtain convergence rates depending on the average delay $\tau_{avg}$. Concurrent to our work, Mishchenko et al. [35] provide a delay-adaptive stepsize scheme and derive convergence guarantees similar to ours. Similar to us, they considered asynchronous SGD with constant stepsizes, but under different assumptions: assuming only Lipschitz-continuity of functions instead of Lipschitz-smoothness. They did not discuss the connection of the number of workers to the average delay. For the heterogeneous case they chose a different approach than ours and provide a delay-adaptive learning rate that converges only to an approximate solution, but allows workers to be arbitrarily long delayed (including the case when some of the workers are never responding).

**Asynchronous federated learning.** In typical federated learning (FL) applications [34], clients or workers frequently have very different computing powers/speed. This makes especially appealing for practitioners to use asynchronous algorithms for FL [47, 37, 54, 8, 53, 23, 10, 22, 52] with many of these works focusing on correcting for unequal participation ratio of different clients [52, 22, 23, 10, 53] by implementing variance reduction techniques on the server. Nguyen et al. [37] introduce the FedBuff algorithm that is very close to the algorithm that we consider in this work and show its practical superiority over classical synchronous FL algorithms.

## 3 Setup

We consider optimization problems where the components of the objective function (i.e. the data for machine learning problems) is distributed across $n$ workers (or clients),

$$\min_{\mathbf{x}\in\mathbb{R}^d}\left[f(\mathbf{x}):=\frac{1}{n}\sum_{i=1}^{n}\left[f_i(\mathbf{x}):=\mathbb{E}_{\xi\sim\mathcal{D}_i}F_i(\mathbf{x},\xi)\right]\right]. \tag{1}$$

Here $f_i\colon\mathbb{R}^d\to\mathbb{R}$ denotes the local objective function that is accessible to the worker $i$, $i\in[n]:=\{1,\dots n\}$. Each $f_i$ is a stochastic function $f_i(\mathbf{x})=\mathbb{E}_{\xi\sim\mathcal{D}_i}F_i(\mathbf{x},\xi)$ and clients can only access stochastic gradients $\nabla F_i(\mathbf{x},\xi)$. This setting covers deterministic optimization if $F_i(\mathbf{x},\xi)=f_i(\mathbf{x})$, $\forall\xi$. It also covers *empirical risk minimization* problems by setting $\mathcal{D}_i$ being a uniform distribution over a local dataset $\{\xi_i^1\dots\xi_i^{m_i}\}$ of size $m_i$. In this case the local functions $f_i$ can be written as finite sums: $f_i(\mathbf{x})=\frac{1}{m_i}\sum_{j=1}^{m_i}F_i(\mathbf{x},\xi_i^j)$.

**Assumptions.** For our convergence analysis we rely on following standard assumptions on the functions $f_i$ and $F_i$:

---

**Algorithm 1** ASYNCHRONOUS SGD

---

**input** Initial value $\mathbf{x}^{(0)} \in \mathbb{R}^d$

1: server selects a set of active workers $\mathcal{C}_0 \subseteq [n]$ and sends them $\mathbf{x}^{(0)}$
2: **for** $t = 0, \dots, T - 1$ **do**
3:     active workers $\mathcal{C}_t$ are computing stochastic gradients in parallel at the assigned points
4:     once a worker $j_t$ finishes compute, it sends $\nabla F(\mathbf{x}^{(t-\tau_t)}, \xi_{t-\tau_t})$ to the server
5:     server updates $\mathbf{x}^{(t+1)} = \mathbf{x}^{(t)} - \eta_t \nabla F(\mathbf{x}^{(t-\tau_t)}, \xi_{t-\tau_t})$
6:     server selects subset $\mathcal{A}_t \subseteq [n]$ of inactive workers, i.e. $(\mathcal{C}_t \backslash \{j_t\}) \cap \mathcal{A}_t = \emptyset$, and sends them $\mathbf{x}^{(t+1)}$
7:     update active worker set $\mathcal{C}_{t+1} = \mathcal{C}_t \backslash \{j_t\} \cup \mathcal{A}_t$
8: **end for**

---

**Assumption 1** (Bounded variance). *We assume that there exists a constant $\sigma \geq 0$ such that*

$$\mathbb{E}_{\xi \sim \mathcal{D}_i} \|\nabla F_i(\mathbf{x}, \xi) - \nabla f_i(\mathbf{x})\| \leq \sigma^2, \qquad\qquad \forall i \in [n], \forall \mathbf{x} \in \mathbb{R}^d. \qquad (2)$$

**Assumption 2** (Bounded function heterogeneity). *We assume that there exists $n$ constants $\zeta_i \geq 0$, $i \in [n]$ such that*

$$\|\nabla f_i(\mathbf{x}) - \nabla f(\mathbf{x})\|_2^2 \leq \zeta_i^2, \quad \forall \mathbf{x} \in \mathbb{R}^d, \qquad \text{and define} \quad \zeta^2 := \tfrac{1}{n} \sum_{i=1}^n \zeta_i^2. \qquad (3)$$

**Assumption 3** ($L$-smoothness). *Each function $f_i \colon \mathbb{R}^d \to \mathbb{R}$, $i \in [n]$ is differentiable and there exists a constant $L \geq 0$ such that*

$$\|\nabla f_i(\mathbf{y}) - \nabla f_i(\mathbf{x})\| \leq L \|\mathbf{x} - \mathbf{y}\|. \qquad\qquad \forall \mathbf{x}, \mathbf{y} \in \mathbb{R}^d. \qquad (4)$$

For only *some* of our results we will assume a uniform bound on the gradient norm:

**Assumption 4** (Bounded gradient). *Each function $f_i \colon \mathbb{R}^d \to \mathbb{R}$, $i \in [n]$ is differentiable and there exists a constant $G \geq 0$ such that*

$$\|\nabla f_i(\mathbf{x})\|_2^2 \leq G^2, \qquad\qquad \forall \mathbf{x} \in \mathbb{R}^d. \qquad (5)$$

## 4 Homogeneous Distributed Setting

We start with an important special case of problem (1) where the objective functions are identical for all workers, i.e. $f_i(\mathbf{x}) \equiv f_j(\mathbf{x})$ for all $i, j \in [n]$, such as in the case of homogeneously (i.i.d.) distributed training data. Consequently, this implies that Assumption 2 holds with $\zeta_i = 0$, $i \in [n]$. Many classical works have focused on asynchronous algorithms under this homogeneous setting (e.g. [6, 48, 1, 20, 45, 28, 46], see the related work for more references). This setting commonly appears in the datacenter setup for distributed training [16], where all nodes (or GPUs) have access to the full dataset or data distribution. Moreover, this special case allows us to present our main ideas in a simplified way, without complicating the presentation due to data heterogeneity. We will later see that most of the results in this section can also be obtained as a corollary of the more general heterogeneous functions case (Section 5) by setting $\zeta_i = 0$ $i \in [n]$.

### 4.1 Algorithm

We consider standard asynchronous SGD (also known as delayed SGD, or SGD with stale updates) as presented in Algorithm 1, see e.g. [6, 48, 1, 20, 45, 28, 46]. First, the server initializes training by selecting an initial active worker set $\mathcal{C}_0$ and assigning $\mathbf{x}^{(0)}$ to these workers. Throughout the algorithm, the active workers compute gradients at their own speed, based on their local data. On line 4, once some worker (which we denote as $j_t$) finishes computing its gradient, it sends the result to the server. On line 5 the server incorporates the received—possibly delayed—gradient, using a stepsize $\eta_t$ that can depend on the gradient delay $\tau_t$. The *gradient delay* $\tau_t$ is defined as the difference between the iteration at which worker $j_t$ started to compute the gradient and the iteration $t$ at which it got applied. We index the stochastic noise of the gradients $\xi_t$ by iteration $t$ to highlight that previous iterates $\mathbf{x}^{(t')}$ for $t' \leq t$ do not depend on this stochastic noise. However, the client selects the data sample $\xi_t$ at iteration $t - \tau_t$ when the computation starts. After that, on lines 6–7 the server selects the new active workers out of the ones that are currently inactive (including worker $j_t$) and assigns them the latest iterate $\mathbf{x}^{(t+1)}$.

In contrast to previous works, we explicitly define the set of workers that are busy with computations at every step $t$ as $\mathcal{C}_t$ (the active workers set). Note that this does not pose any restrictions. A main advantage of allowing the sets $\mathcal{C}_t$ to be different at every step $t$ lies in the possibility to also cover mini-batch SGD as a special case, which we discuss in Example 2. Our theoretical results depend on the size of these sets $\mathcal{C}_t$, a.k.a. the *concurrency*.

**Definition 1** (Concurrency). *The* concurrency $\tau_C^{(t)}$ *at step $t$ is defined as the size of the active worker set $\mathcal{C}_t$, i.e. $\tau_C^{(t)} = |\mathcal{C}_t|$. We also define the maximum and average concurrency as*

$$\tau_C = \max_t \{\tau_C^{(t)}\}, \qquad\qquad \bar{\tau}_C = \tfrac{1}{T+1} \sum_{t=0}^{T} \tau_C^{(t)}.$$

Note that in many practical scenarios, we have a *constant concurrency* of $n$ over time, meaning that all $n$ workers are active at every step, and thus $\tau_C = \bar{\tau}_C = n$.

We discuss two important practical examples that fit into our Algorithm 1:

**Example 2** (Mini-batch SGD). *Mini-batch SGD with batch size $n$ can be seen as a special case of Algorithm 1, as follows: The server (i) in line 1 selects all $n$ workers, $\mathcal{C}_0 = [n]$; (ii) in line 6 does not select new workers while the gradients from the same batch have not been fully applied yet, i.e. $\mathcal{A}_t = \emptyset$ if $t \bmod n \neq 0$; (iii) in line 6 selects $\mathcal{A}_t = [n]$ if $t \bmod n = 0$ to start a new batch.*

**Example 3** (Asynchronous SGD with maximum concurrency). *In practical implementations one should always aim to utilize all resources available and thus (i) in line 1 select all available workers $\mathcal{C}_0 = [n]$; (ii) in line 6 select to re-assign the worker that finished its computations $\mathcal{A}_t = \{j_t\}$ so that workers are always kept busy with jobs.*

### 4.2 Theoretical analysis: Constant stepsizes

We first formally define the average and maximum delays.

**Definition 4** (Average and maximum delays). *Let $\{\tau_t\}_{t=0}^{T-1}$ be the delays of the applied gradients in Algorithm 1. We define $\{\tau_i^{\mathcal{C}_T}\}_{i \in \mathcal{C}_T \setminus \{j_T\}}$ as the delays of gradients which are in flight at time $T$, that is they have remained unapplied at the last step. Each $\tau_i^{\mathcal{C}_T}$ is equal to the difference between the last iteration $T$ and the iteration at which worker $i$ started to compute its last gradient. We then define the average and the maximum delays as*

$$\tau_{avg} = \frac{1}{T + |\mathcal{C}_T| - 1} \left( \sum_{t=0}^{T-1} \tau_t + \sum_{i \in \mathcal{C}_T \setminus \{j_T\}} \tau_i^{\mathcal{C}_T} \right), \quad \tau_{\max} = \max \left\{ \max_{t=1,\dots T-1} \tau_t, \max_{i \in \mathcal{C}_T \setminus \{j_T\}} \tau_i^{\mathcal{C}_T} \right\}. \quad (6)$$

We further provide a key observation on the connection between the average delay and the average concurrency. This observation, is one of the essential elements for achieving an improved analysis.

**Remark 5** (Key Observation). *In Algorithm 1 the average concurrency $\bar{\tau}_C$ is connected to the average delay $\tau_{avg}$ as*

$$\tau_{avg} = \frac{T+1}{T + |\mathcal{C}_T| - 1} \bar{\tau}_C \;\overset{T > |\mathcal{C}_T|}{=}\; \Theta(\bar{\tau}_C). \quad (7)$$

We explain this observation on a simple example. Assume that the concurrency is constant at every step ($\tau_C = \bar{\tau}_C$), and that all workers except one are responding very rarely. Then on steps 4–5 of Algorithm 1 only this one responding worker would mostly participate. This means that for this one worker the delay $\tau_t$ would be frequently equal to zero, and the overall average delay will be small.

Next, we provide our theoretical results. We first focus on the Asynchronous SGD Algorithm 1 under constant stepsizes, i.e. $\eta_t \equiv \eta$. This setting was studied in many works such as [1, 20, 6, 28, 48].

**Theorem 6** (Constant stepsizes). *Under Assumptions 1, 3, there exists a constant stepsize $\eta_t \equiv \eta$ such that for Algorithm 1 it holds that $\frac{1}{T+1} \sum_{t=0}^{T} \left\| \nabla f(\mathbf{x}^{(t)}) \right\|_2^2 \leq \varepsilon$ after*

$$\mathcal{O}\left( \frac{\sigma^2}{\varepsilon^2} + \frac{\sqrt{\tau_C \tau_{\max}}}{\varepsilon} \right) \qquad\qquad \text{iterations.} \quad (8)$$

*If we additionally assume bounded gradient Assumption 4, then $\frac{1}{\sum_{t=0}^{T} |\mathcal{A}_t|} \sum_{t=0}^{T} |\mathcal{A}_t| \left\| \nabla f(\mathbf{x}^{(t)}) \right\|_2^2 \leq \varepsilon$ after*

$$\mathcal{O}\left( \frac{\sigma^2}{\varepsilon^2} + \frac{\tau_C G}{\varepsilon^{3/2}} + \frac{\tau_C}{\varepsilon} \right) \qquad\qquad \text{iterations.} \quad (9)$$

Under constant concurrency, we can directly connect $\tau_C$ to the average delay $\tau_{avg}$ due to Remark 5. We highlight again that in practice, to get the best utilization of the available resources, practical implementations choose the maximum concurrency possible, which is equal to $n$.

**Corollary 7.** *If in Algorithm 1 the concurrency is constant at every step (thus $\tau_C = \bar{\tau}_C$), then under the same conditions as in Theorem 6 the convergence rate of Algorithm 1 is*

$$\mathcal{O}\left(\frac{\sigma^2}{\varepsilon^2} + \frac{\sqrt{\tau_{avg}\tau_{\max}}}{\varepsilon}\right) \qquad and \qquad \mathcal{O}\left(\frac{\sigma^2}{\varepsilon^2} + \frac{\tau_{avg}G}{\varepsilon^{3/2}} + \frac{\tau_{avg}}{\varepsilon}\right) \qquad (10)$$

*for the case without and with bounded gradients (Assumption 4) respectively.*

The previously best known convergence rate for Asynchronous SGD (Algorithm 1) under constant stepsizes was given in [48] and is equal to $\mathcal{O}\left(\frac{\sigma^2}{\varepsilon^2} + \frac{\tau_{\max}}{\varepsilon}\right)$. In our theorem we improved the delay dependence from $\tau_{\max}$ to $\sqrt{\tau_{avg}\tau_{\max}}$ in the last term *without any change in the algorithm*, only by taking into account concurrency that is usually fixed in practical implementations anyways. No other work previously made an assumption on the number of computing workers in their theoretical analysis. $\sqrt{\tau_{avg}\tau_{\max}}$ could be much smaller than $\tau_{\max}$ in the presence of rare straggler devices. With an additional assumption of bounded gradients, the dependence on the maximum delay can be completely removed.

### 4.3 Theoretical analysis: Delay-adaptive stepsizes

In many cases, the bounded gradient Assumption 4 is unrealistic [38], meaning that the gradient bound $G$ is often large and thus the rate (9) is loose. In this section we show that by weighting the stepsize down for the gradients that have a large delay, once can remove the dependence on the maximum delay $\tau_{\max}$ without assuming bounded gradients (Assump. 4).

**Theorem 8** (Delay-adaptive stepsizes). *There exist a parameter $\eta \leq \frac{1}{4L}$ such that if we set the stepsizes in Algorithm 1 dependent on the delays as*

$$\eta_t = \begin{cases} \eta & \tau_t \leq 2\tau_C, \\ < \min\{\eta, \frac{1}{4L\tau_t}\} & \tau_t > 2\tau_C, \end{cases} \qquad (11)$$

*then for Algorithm 1, under Assumptions 1, 3 it holds that $\frac{1}{\sum_{t=0}^{T}\eta_t}\sum_{t=0}^{T}\eta_t \left\|\nabla f(\mathbf{x}^{(t)})\right\|_2^2 \leq \varepsilon$ after*

$$\mathcal{O}\left(\frac{\sigma^2}{\varepsilon^2} + \frac{\tau_C}{\varepsilon}\right) \qquad\qquad iterations. \qquad (12)$$

In our theorem, the stepsize $\eta_t$ in the case of large delays $\tau_t > 2\tau_C$ can be an arbitrary value between $0$ and $\min\{\eta, \frac{1}{4L\tau_t}\}$. Setting the stepsize $\eta_t \equiv 0$ is equivalent to dropping these gradients.

*Proof sketch of Theorem 8.* We give the intuitive proof sketch for the case when we drop gradients with $\tau_t > 2\tau_C$ and we deal with the general case in the Appendix. We know that $\tau_{avg} \approx \bar{\tau}_C \leq \tau_C$ from Remark 5. It also holds that the number of gradients that have delay larger than the two times the average delay $2\tau_{avg}$ is smaller than half of all the gradients ($\leq \frac{T}{2}$) because delays are bounded below by zero ($\tau_t \geq 0 \;\forall t$). Thus, dropping the gradients with the delay $\tau_t > 2\tau_C$, or equivalently setting their stepsize $\eta_t \equiv 0$, will degrade the convergence rate at most by half, while the maximum delay among the applied ones now is equal to $2\tau_C$. Thus we can apply the result from [48] with $\tau_{max} = 2\tau_C$. $\qquad\square$

**Corollary 9.** *If in Algorithm 1 the concurrency is constant at every step (thus $\tau_C = \bar{\tau}_C$), then under the same conditions as in Theorem 8 the convergence rate of Algorithm 1 is*

$$\mathcal{O}\left(\frac{\sigma^2}{\varepsilon^2} + \frac{\tau_{avg}}{\varepsilon}\right). \qquad (13)$$

### 4.4 Discussion

**Comparison to synchronous optimization.** Mini-batch SGD with batch size $n$ has the same degree of parallelism as Algorithm 1 with constant concurrency $n$, i.e. it has $n$ workers computing gradients in parallel. Mini-batch SGD needs $\mathcal{O}\left(\frac{\sigma^2}{n\varepsilon^2} + \frac{1}{\varepsilon}\right)$ [21] batches of gradients to reach an $\varepsilon$-stationary point, and thus needs $\mathcal{O}\left(\frac{\sigma^2}{\varepsilon^2} + \frac{n}{\varepsilon}\right)$ gradients, as the batch-size is equal to $n$. On the

contrary, asynchronous SGD Algorithm 1 with stepsizes chosen as in (11) achieves exactly the same rate (13) since $\tau_{avg} = \tau_C = n$, while its expected per-iteration time is *faster* than that of mini-batch SGD, as no workers have to wait for others. Thus, our result shows that asynchronous SGD is always faster than mini-batch SGD *regardless of the delay pattern*. A small note that in our reasoning we implicitly assumed that the sever can perform its operations in negligible time.

**Tuning the stepsize.** It is worth noting that our stepsize rule (11) does not introduce any additional hyperparameters to tune compared to the constant stepsize case or to synchronous SGD. $\tau_C$ is usually known and can be easily controlled by the server, especially in the practical constant concurrency case. Thus, to implement such a stepsize rule (11) one needs to tune only stepsize $\eta$, and in case of $\tau_t > 2\tau_C$ set stepsize $\eta_t \leq \frac{\eta}{\tau_t}$.

**Average v.s. maximum delay.** In a homogeneous environment when every worker computes gradients with same speed during the whole training, the average and maximum delays would be almost equal. However, occasional straggler devices will usually be present. In this case the maximum delay is much larger than the average delay.

Consider a simple example with $n = 2$ workers, where the first worker computes gradients very fast, while the second worker returns its gradient only at the end of the training at the last iteration $T$. In this case the average delay $\tau_{avg} = 2$ is a small constant, while the maximum delay $\tau_{\max} = T$. In this case the rate depending only on the maximum delay $\tau_{\max}$ would guarantee convergence only up to a constant accuracy $\varepsilon = \mathcal{O}(1)$. While both rates with $\sqrt{\tau_{\max}\tau_{avg}}$ and with $\tau_{avg}$ guarantee convergence up to an arbitrary small accuracy.

**Comparison to other methods.** Cohen et al. [15] recently proposed the PickySGD algorithm that achieves a convergence rate of $\mathcal{O}\left(\frac{\sigma^2}{\varepsilon^2} + \frac{\tau_{avg}}{\varepsilon}\right)$ (same as (13)). Their algorithm discards gradients based on the distance between the current point and the delayed one $\left\|\mathbf{x}^{(t)} - \mathbf{x}^{(t-\tau_t)}\right\|$. The disadvantage of their method is that it requires sending points $\mathbf{x}^{(t-\tau_t)}$ along with the gradients thus incurring twice more communications at every step. Their method also requires tuning an extra hyperparameter. In this work we achieve the same convergence rate with a much simpler method that does not require any additional communications nor additional tuning compared to synchronous SGD.

[9] also recently proposed the delay-adaptive algorithm with convergence rate depending on the average delay $\tau_{avg}$ for the convex and strongly convex cases. Although, our convergence rates are for the non-convex case and are not directly comparable to theirs, we highlight some key differences in their analysis. First, their convergence rate depends not only on $\tau_{avg}$ but also on the variance $\sigma_\tau$ of the delays, which can degrade with the maximum delay. Second, they require the bounded gradient Assumption 4. In Theorem 6 we show that under Assumption 4 no modifications to the algorithm are needed to completely remove the dependence on the maximum delay $\tau_{\max}$ (9).

**Tightness.** As we explained in Example 2, mini-batch SGD is covered by Algorithm 1. We know that mini-batch SGD convergence is lower bounded by $\Theta\left(\frac{\sigma^2}{n\varepsilon^2} + \frac{1}{\varepsilon}\right)$ [5] in terms of batches processed and thus by $\Theta\left(\frac{\sigma^2}{\varepsilon^2} + \frac{n}{\varepsilon}\right)$ in terms of the gradients computed. Our convergence rate given in Theorem 6 *coincides with this lower bound* as in this case concurrency $\tau_C = n$, $\tau_{avg} = \bar{\tau}_C = \frac{n}{2}$.

## 5 Heterogeneous Distributed Setting

In this section we consider more general problems of the form (1) where the functions $f_i$ are different on different workers. This setting is motivated for example by federated learning [34, 24], where every worker (client) possesses its own private data, possibly coming from a different data distribution, and thus has its own different local objective function $f_i$.

The setting here is therefore more general than the one considered in previous Section 4, and we will see that some of the results (with the constant stepsizes) in the homogeneous case follow as a special case of the more general results we present in this section.

### 5.1 Algorithm

We consider asynchronous SGD as given in Algorithm 2. Close variants of this algorithm were studied in several prior works [37, 47]. In order to simplify the presentation, we consider that concurrency is constant over time (and thus $\tau_C = \bar{\tau}_C$ in Definition 1). In order to allow for client subsampling

often implemented in practical federated learning applications, we allow the concurrency $\tau_C$ to be smaller than overall number of workers $n$. The same concurrency model was recently considered in the practical FedBuff algorithm [37].

---

**Algorithm 2** ASYNCHRONOUS SGD with concurrency $\tau_C$

---

**input** Initial value $\mathbf{x}^{(0)} \in \mathbb{R}^d$, $n$ clients, concurrency $\tau_C$

    **Server:**
1: server selects *uniformly at random* a set of active clients $\mathcal{C}_0$ of size $\tau_C$ and sends them $\mathbf{x}^{(0)}$
2: **for** $t = 0, \ldots, T - 1$ **do**
3:     active clients $\mathcal{C}_t$ are computing stochastic gradients in parallel at the assigned points
4:     once some client $j_t$ finishes compute, it sends $\nabla F_{j_t}(\mathbf{x}^{(t-\tau_t)}, \xi_{t-\tau_t})$ to the server
5:     server updates $\mathbf{x}^{(t+1)} = \mathbf{x}^{(t)} - \eta_t \nabla F_{j_t}(\mathbf{x}^{(t-\tau_t)}, \xi_{t-\tau_t})$
6:     sever selects a new client $k_t \sim \text{Uniform}[1, n]$ and sends it $\mathbf{x}^{(t+1)}$
7:     update the active worker multiset $\mathcal{C}_{t+1} = \mathcal{C}_t \backslash \{j_t\} \cup \{k_t\}$
8: **end for**

---

The algorithm is very similar to the homogeneous Algorithm 1 with two key differences: at line 6, the server selects clients *out of all clients*, and does so *uniformly at random*, regardless of the current active worker set $\mathcal{C}_t$. This means that the same client can get sampled several times, even if it didn't finish its previous job(s) yet (thus $\mathcal{C}_t$ is a multiset). In this case, the assigned jobs would just pile up on this client.

### 5.2 Theoretical analysis

We first note that our key observation on the delays (Remark 5) holds for Algorithm 2 as well. Moreover, as we have a constant concurrency $\tau_C$ at every step, $\tau_{avg} = \Theta(\tau_C)$.

**Definition 10.** *Denote a (possibly empty) set $\{\tau_k^{C_T, i}\}_k$ to be the set of delays from gradients of the client $i$ that are left unapplied at the last iteration of the Algorithm 2.*

*We define the average delay of a client $i$ as*

$$\tau_{avg}^i = \frac{1}{T_i} \left( \sum_{t \,:\, j_t = i} \tau_t + \sum_k \tau_k^{C_T, i} \right)$$

*where $T_i$ is the number of times that client $i$ was sampled during lines 1 and 6 of Algorithm 2.*

**Assumption 5.** *The average delay $\tau_{avg}^i$ is independent from the number of times $T_i$ that client $i$ was sampled.*

**Theorem 11** (constant stepsizes). *Under Assumptions 1, 2, 3, 5 there exists a constant stepsize $\eta_t \equiv \eta$ such that for Algorithm 2 it holds that $\frac{1}{T+1} \sum_{t=0}^T \left\| \nabla f(\mathbf{x}^{(t)}) \right\|_2^2 \leq \varepsilon$ after*

$$\mathcal{O}\left( \frac{\sigma^2}{\varepsilon^2} + \frac{\zeta^2}{\varepsilon^2} + \frac{\sqrt{\tau_{avg} \frac{1}{n} \sum_{i=1}^n \zeta_i^2 \tau_{avg}^i}}{\varepsilon^{\frac{3}{2}}} + \frac{\sqrt{\tau_{avg} \tau_{\max}}}{\varepsilon} \right) \qquad \text{iterations,} \qquad (14)$$

*Under Assumptions 1, 2, 3 and additional bounded gradient Assumption 4, it holds that $\frac{1}{T+1} \sum_{t=0}^T \left\| \nabla f(\mathbf{x}^{(t)}) \right\|_2^2 \leq \varepsilon$ after*

$$\mathcal{O}\left( \frac{\sigma^2}{\varepsilon^2} + \frac{\zeta^2}{\varepsilon^2} + \frac{\tau_{avg} G}{\varepsilon^{\frac{3}{2}}} + \frac{\tau_{avg}}{\varepsilon} \right) \qquad \text{iterations.} \qquad (15)$$

We note that the leading $\frac{1}{\varepsilon^2}$ term is affected by heterogeneity $\zeta^2$ because at every step we apply gradient from only one client. This term is usually present in the federated learning algorithms with client subsampling see e.g. [25].

### 5.3 Discussion

**Comparison to other works.** The recent FedBuff algorithm [37] is similar to our Algorithm 2. Their algorithm allows clients to perform several local steps and the server to wait for more than 1

client to finish compute (aka buffering), which we did not include for simplicity as these aspects are orthogonal to the effect of delays.

Disregarding these two orthogonal changes, the FedBuff algorithm is almost equivalent to our Algorithm 2 with a key difference: they assume that the client $j_t$ that finishes computation at every step comes from the uniform distribution over all the clients. This is unrealistic to assume in practice because the server cannot control which clients finish computations at every step. In Algorithm 2 we have the more realistic assumption only on the sampling process of the new clients (on line 6) that *can be controlled* by the server. This reflects practical client sampling in federated learning.

The convergence rate of FedBuff [37] under the bounded gradient assumption is $\mathcal{O}\left(\frac{\sigma^2}{\varepsilon^2} + \frac{\zeta^2}{\varepsilon^2} + \frac{(\zeta^2+1)\tau_{\max}G^2}{\varepsilon}\right)$. In contrast, in Theorem 11 we completely remove the dependence on the maximum delay $\tau_{\max}$ under bounded gradients (as in Equation (15)).

**Delays.** We note that for Theorem 11 we did not impose any assumption on the delays. Thus, our result allows clients and the delays on these clients to be dependent, meaning that some of the clients could be systematically slower than others. Interestingly, the middle heterogeneity term (the term with $\zeta_i$) is not affected by the maximum delay at all, but is affected by the average delay within each individual client. If all the heterogeneity parameters are equal, i.e. $\zeta_i = \zeta_j, \forall i, j$, then the middle term will be affected only by the overall average delay $\tau_{avg}$.

**Gradient clipping.** Practical implementations of FL algorithms usually apply clipping to the gradients in order to guarantee differential privacy [24]. This automatically bounds the norm of all applied gradients, making the the constant $G^2$ in Assumption 4 small. Although we do not provide formal convergence guarantees of asynchronous SGD with gradient clipping, we envision that its convergence rate would depend only on the average delay, similar to the bounded gradient case (9), thus making the algorithm robust to stragglers.

**Delay-adaptive stepsizes.** For homogeneous functions we have shown that delay-adaptive stepsizes result in a convergence rate dependent only on the average delay $\tau_{avg}$ without assuming bounded gradients (as in Equation (11)). However in the heterogeneous case this is not so straightforward. Delay-adaptive learning rate schemes will introduce a bias towards the clients that compute quickly, and Algorithm 2 would converge to the wrong objective.

It is interesting to note that current popular schemes implemented in practice for FL over-selects the clients at every iteration [12]. The server waits only for some percentage (e.g. 80%) of sampled clients and discards the rest. Such a scheme also introduces a bias towards fast workers. A delay-adaptive learning rate scheme is expected to introduce less bias as the gradients are still applied but with the smaller weight. We leave this question for future practical investigations, as it is not the focus of our current work.

**Independent delays.** If the delays and the clients are independent (e.g. coming from the same distribution for all of the clients), then the convergence rate of Algorithm 2 will simplify to $\mathcal{O}\left(\frac{\sigma^2}{\varepsilon^2} + \frac{\zeta\tau_{avg}}{\varepsilon^{\frac{3}{2}}} + \frac{\sqrt{\tau_{avg}\tau_{\max}}}{\varepsilon}\right)$ (without needing bounded gradient assumption). In this case it is also possible to use delay-adaptive stepsizes (similar to Theorem 8) to completely remove the dependence on the maximum delays $\tau_{\max}$ without assuming bounded gradients.

**Extensions.** We can extend the Algorithm 2 and our theoretical analysis to allow clients to perform several local steps, before sending back the change in $\mathbf{x}$. We can also extend Algorithm 2 to allow the server to wait for the first $K$ clients to finish computations rather than just one, similar to [37]. These extensions are straightforward and we excluded them here for simplicity of presentation.

Finally, we can also extend Algorithm 2 to sample new clients as soon as some previous client finished compute, without waiting for the server update on the line 5.

## 6 Experiments

In this section we aim to empirically evaluate the effectiveness of our proposed delay-adaptive stepsizes (11) in the homogeneous case. We use w1a dataset from LIBSVM data library, and evaluate performance on logistic regression function defined as $f(\mathbf{x}) = \frac{1}{m}\sum_{j=1}^{m} \log(1 + \exp(-b_j \mathbf{a}_j^\top \mathbf{x})) +$

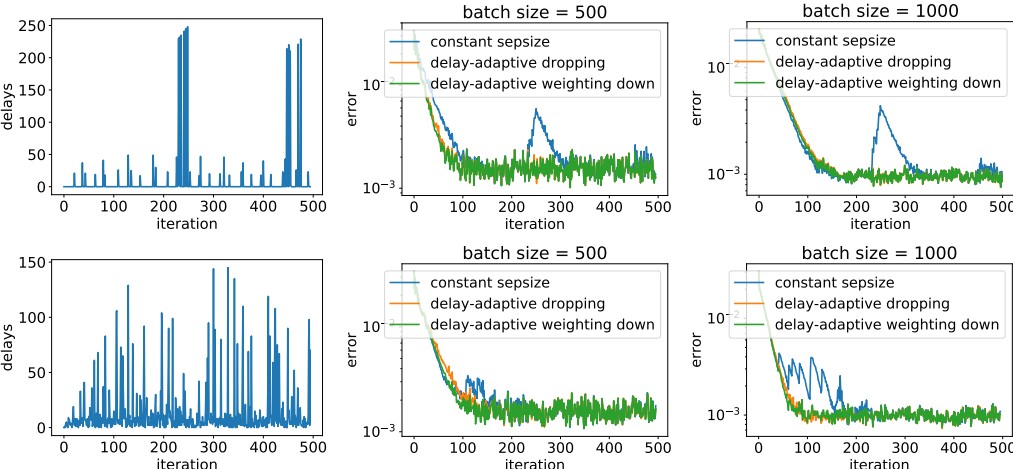

Figure 1: Comparison of three different stepsize strategies in asynchronous SGD. (left) delay pattern, (middle) convergence curve with batch size 500, (right) convergence curves with batch size 1000. Both delay-adaptive stepsize schemes are more robust to occasional large delays.

$\frac{1}{2m}\|\mathbf{x}\|^2$, where $\mathbf{a}_j \in \mathbb{R}^d$ and $b_j \in \{-1, 1\}$ are data samples. w1a dataset has $m = 2477$ examples and dimension $d = 300$. We use fixed concurrency with $n = 10$ workers.

We compare the following three stepsize strategies:

- constant stepsizes $\eta_t \equiv \eta$
- delay-adaptive stepsizes with dropping large-delayed gradients

$$\eta_t = \begin{cases} \eta & \tau_t \le 2n, \\ 0 & \tau_t > 2n, \end{cases}$$

- delay-adaptive stepsizes with weighting down large-delayed gradients

$$\eta_t = \begin{cases} \eta & \tau_t \le 2n, \\ \eta\frac{\tau_C}{\tau_t} & \tau_t > 2n, \end{cases}$$

Note that the two last stepsize schemes satisfy the conditions in (11). For each of the strategies we tune only one parameter $\eta$ separately from the logarithmic grid and ensure that the found stepsize is not on the edge of this grid.

Figure 1 presents the results for two different delay patterns. We can see that both of the delay-adaptive stepsizes have comparable performance and are more robust to the large delays than the constant stepsize.

## 7 Conclusion

In this paper we study the asynchronous SGD algorithm both in homogeneous and heterogeneous settings. By leveraging the notion of concurrency—the number of workers that compute gradients in parallel—we show a much faster convergence rate for asynchronous SGD, improving the dependence on the maximum delay $\tau_{\max}$ over prior works, for both homogeneous and heterogeneous objectives. Our proof technique also allows to design a simple delay-adaptive stepsize rule (11) that attains a convergence rate depending only on the average delay $\tau_{avg}$ that neither requires any additional tuning, nor additional communication. Our techniques allows us to demonstrate that *asynchronous SGD is faster than mini-batch SGD for any delay pattern*.

## Acknowledgments

AK was supported by a Google PhD Fellowship. We thank Brendan McMahan, Thijs Vogels, Hadrien Hendrikx, Aditya Vardhan Varre and Maria-Luiza Vladarean for useful discussions and their feedback on the manuscript.

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
