# A Proofs

In this section we provide the proofs of all the theoretical results stated in the main paper.

## A.1 Proof of Remark 5

First, we prove our key observation given in Remark 5.

**Remark 5** (Key Observation). *In Algorithm 1 the average concurrency $\bar{\tau}_C$ is connected to the average delay $\tau_{avg}$ as*

$$\tau_{avg} = \frac{T+1}{T+|C_T|-1}\bar{\tau}_C \stackrel{T>|C_T|}{=} \Theta(\bar{\tau}_C) . \tag{7}$$

*Proof.* Define $\{\tau_i^{C_t}\}_{i\in C_t}$ as the set of delays of the gradients that are left in the active worker set before iteration $t$ is performed, i.e. each $\tau_i^{C_t}$ is equal to the difference between the current iteration $t$ and the iteration at which worker $i$ started to compute its current gradient for $t > 0$, and $\tau_i^{C_0} = 1$ for all $i \in C_0$, that is the initial set of active workers. For simplicity we denote

$$\tau_{sum}^{\text{active},t} := \sum_{i\in C_t} \tau_i^{C_t} .$$

We also define $\tau_{sum}^{\text{applied},t}$ as the sum of all delays of gradients applied before iteration $t$ is performed, i.e.

$$\tau_{sum}^{\text{applied},t} := \sum_{j=0}^{t-1} \tau_j .$$

At the zero-th iteration we have that

$$\tau_{sum}^{\text{applied},0} = 0, \qquad\qquad \tau_{sum}^{\text{active},0} = \tau_C^{(0)}, \tag{16}$$

as no gradients were applied yet.

We claim that

$$\tau_{sum}^{\text{applied},t+1} + \tau_{sum}^{\text{active},t+1} = \tau_{sum}^{\text{applied},t} + \tau_{sum}^{\text{active},t} + \tau_C^{(t+1)} . \tag{17}$$

Indeed, one of the gradients from $C_t$ got applied and its delay moved from $\tau_{sum}^{\text{active},t}$ to $\tau_{sum}^{\text{applied},t+1}$. The newly selected active workers in line 6 of Algorithm 1 have delay zero, as they just started their computations in this step. And all of the current active workers in $C_{t+1}$ (of size $|C_{t+1}| = \tau_C^{(t+1)}$) got an increase by 1 due to increase of the iteration count from $t$ to $t+1$.

Using the initial conditions (16) and (17) we can conclude that

$$\tau_{sum}^{\text{applied},T} + \tau_{sum}^{\text{active},T} = \sum_{t=0}^{T} \tau_C^{(t)} = (T+1)\bar{\tau}_C .$$

Note that the left hand side is exactly equal to $(T+|C_T|-1)\tau_{avg}$ from our Definition 4. Thus,

$$\tau_{avg} = \frac{T+1}{T+|C_T|-1}\bar{\tau}_C = \mathcal{O}\left(\bar{\tau}_C\right) ,$$

where the last equality holds if $T > |C_T|$. □

## A.2 Useful inequalities

**Lemma 12.** *For an arbitrary set of $n$ vectors $\{\mathbf{a}_i\}_{i=1}^n$, $\mathbf{a}_i \in \mathbb{R}^d$*

$$\left\|\sum_{i=1}^{n} \mathbf{a}_i\right\|^2 \leq n \sum_{i=1}^{n} \|\mathbf{a}_i\|^2 . \tag{18}$$

## A.3 Proof of Theorems 6, (8) and 8

We first recall both of the theorems

**Theorem 6** (Constant stepsizes). *Under Assumptions 1, 3, there exists a constant stepsize $\eta_t \equiv \eta$ such that for Algorithm 1 it holds that $\frac{1}{T+1}\sum_{t=0}^{T} \left\| \nabla f(\mathbf{x}^{(t)}) \right\|_2^2 \leq \varepsilon$ after*

$$\mathcal{O}\left( \frac{\sigma^2}{\varepsilon^2} + \frac{\sqrt{\tau_C \tau_{\max}}}{\varepsilon} \right) \qquad\qquad \textit{iterations.} \qquad (8)$$

*If we additionally assume bounded gradient Assumption 4, then $\frac{1}{\sum_{t=0}^{T}|\mathcal{A}_t|}\sum_{t=0}^{T} |\mathcal{A}_t| \left\| \nabla f(\mathbf{x}^{(t)}) \right\|_2^2 \leq \varepsilon$ after*

$$\mathcal{O}\left( \frac{\sigma^2}{\varepsilon^2} + \frac{\tau_C G}{\varepsilon^{3/2}} + \frac{\tau_C}{\varepsilon} \right) \qquad\qquad \textit{iterations.} \qquad (9)$$

**Theorem 8** (Delay-adaptive stepsizes). *There exist a parameter $\eta \leq \frac{1}{4L}$ such that if we set the stepsizes in Algorithm 1 dependent on the delays as*

$$\eta_t = \begin{cases} \eta & \tau_t \leq 2\tau_C, \\ < \min\{\eta, \frac{1}{4L\tau_t}\} & \tau_t > 2\tau_C, \end{cases} \qquad\qquad (11)$$

*then for Algorithm 1, under Assumptions 1, 3 it holds that $\frac{1}{\sum_{t=0}^{T}\eta_t}\sum_{t=0}^{T} \eta_t \left\| \nabla f(\mathbf{x}^{(t)}) \right\|_2^2 \leq \varepsilon$ after*

$$\mathcal{O}\left( \frac{\sigma^2}{\varepsilon^2} + \frac{\tau_C}{\varepsilon} \right) \qquad\qquad \textit{iterations.} \qquad (12)$$

We first give a common lemma that will be used in the proofs for both of the theorems.

**Lemma 13** (Descent Lemma). *Under Assumptions 1 and 3, if in Algorithm 1 the stepsize $\eta_t < \frac{1}{2L}$ then it holds that*

$$\mathbb{E}_{t+1}\, f(\mathbf{x}^{(t+1)}) \leq f(\mathbf{x}^{(t)}) - \frac{\eta_t}{2} \left\| \nabla f(\mathbf{x}^{(t)}) \right\|_2^2 - \frac{\eta_t}{4} \left\| \nabla f(\mathbf{x}^{(t-\tau_t)}) \right\|^2 + L\eta_t^2 \sigma^2 + \frac{\eta_t L^2}{2} \left\| \mathbf{x}^{(t)} - \mathbf{x}^{(t-\tau_t)} \right\|_2^2, .$$

*Proof.* Because the function $f$ is $L$-smooth, we have

$$\mathbb{E}_{t+1}\, f(\mathbf{x}^{(t+1)}) = \mathbb{E}_{t+1}\, f\left( \mathbf{x}^{(t)} - \eta_t \nabla F(\mathbf{x}^{(t-\tau_t)}, \xi_{t-\tau_t}) \right)$$

$$\leq f(\mathbf{x}^{(t)}) - \eta_t \underbrace{\mathbb{E}_{t+1} \left\langle \nabla f(\mathbf{x}^{(t)}), \nabla F(\mathbf{x}^{(t-\tau_t)}, \xi_{t-\tau_t}) \right\rangle}_{=:T_1} + \mathbb{E}_{t+1}\, \frac{L}{2}\eta_t^2 \underbrace{\left\| \nabla F(\mathbf{x}^{(t-\tau_t)}, \xi_{t-\tau_t}) \right\|_2^2}_{=:T_2}$$

We first estimate the second term as

$$T_1 = -\eta_t \left\langle \nabla f(\mathbf{x}^{(t)}), \nabla f(\mathbf{x}^{(t-\tau_t)}) \right\rangle = -\frac{\eta_t}{2} \left\| \nabla f(\mathbf{x}^{(t)}) \right\|^2 - \frac{\eta_t}{2} \left\| \nabla f(\mathbf{x}^{(t-\tau_t)}) \right\|^2 + \frac{\eta_t}{2} \left\| \nabla f(\mathbf{x}^{(t)}) - \nabla f(\mathbf{x}^{(t-\tau_t)}) \right\|^2$$

For the last term, we add and subtract $\nabla f(\mathbf{x}^{(t-\tau_t)})$, and use that $\mathbb{E}_{t+1}\, \nabla F(\mathbf{x}^{(t-\tau_t)}, \xi_{t-\tau_t}) - \nabla f(\mathbf{x}^{(t-\tau_t)}) = 0$

$$T_2 = \mathbb{E}_{t+1} \left\| \nabla F(\mathbf{x}^{(t-\tau_t)}, \xi_{t-\tau_t}) - \nabla f(\mathbf{x}^{(t-\tau_t)}) \right\|_2^2 + \left\| \nabla f(\mathbf{x}^{(t-\tau_t)}) \right\|_2^2$$

$$\overset{(2)}{\leq} \sigma^2 + \left\| \nabla f(\mathbf{x}^{(t-\tau_t)}) \right\|_2^2 .$$

Combining this together and using $L$-smoothness to estimate $\left\| \nabla f(\mathbf{x}^{(t)}) - \nabla f(\mathbf{x}^{(t-\tau_t)}) \right\|_2^2$,

$$\mathbb{E}_{t+1}\, f(\mathbf{x}^{(t+1)}) \leq f(\mathbf{x}^{(t)}) - \eta_t \left\| \nabla f(\mathbf{x}^{(t)}) \right\|_2^2 - \frac{\eta_t}{2}(1 - L\eta_t) \left\| \nabla f(\mathbf{x}^{(t-\tau_t)}) \right\|_2^2 + \frac{\eta_t L^2}{2} \left\| \mathbf{x}^{(t)} - \mathbf{x}^{(t-\tau_t)} \right\|_2^2$$

$$+ L\eta_t^2 \sigma^2 .$$

Applying $\eta < \frac{1}{2L}$ we get statement of the lemma. $\qquad\square$

### A.3.1 Proof of Theorem 6, convergence rate (8)

**Lemma 14** (Estimation of the residual). *Under Assumptions 1 and 3, the iterates of Algorithm 1 with the constant stepsize $\eta_t \equiv \eta$ with $\eta \leq \frac{1}{2L\sqrt{\tau_{\max}\tau_C}}$ satisfy*

$$\frac{1}{T+1}\sum_{t=0}^{T}\mathbb{E}\left\|\mathbf{x}^{(t)}-\mathbf{x}^{(t-\tau_t)}\right\|_2^2 \leq \frac{1}{4L^2(T+1)}\sum_{t=0}^{T}\mathbb{E}\left\|\nabla f(\mathbf{x}^{(t-\tau_t)})\right\|^2 + \frac{\sigma^2\eta}{2L}\,.$$

*Proof.* We start with unrolling the difference and use that $\mathbb{E}\,\nabla F(\mathbf{x}^{(j-\tau_j)}, \xi^{(j-\tau_j)}) = \nabla f(\mathbf{x}^{(j-\tau_j)})$.

$$\mathbb{E}\left\|\mathbf{x}^{(t)}-\mathbf{x}^{(t-\tau_t)}\right\|_2^2 = \mathbb{E}\left\|\sum_{j=t-\tau_t}^{t-1}\eta\nabla F(\mathbf{x}^{(j-\tau_j)}, \xi_{j-\tau_j})\right\|^2 \overset{(5)}{\leq} \mathbb{E}\left\|\sum_{j=t-\tau_t}^{t-1}\eta\nabla f(\mathbf{x}^{(j-\tau_j)})\right\|^2 + \tau_t\eta^2\sigma^2$$

$$\overset{(18)}{\leq} \tau_t\,\mathbb{E}\sum_{j=t-\tau_t}^{t-1}\eta^2\left\|\nabla f(\mathbf{x}^{(j-\tau_j)})\right\|^2 + \tau_t\eta^2\sigma^2\,.$$

Using that $\eta \leq \frac{1}{2L\sqrt{\tau_{\max}\tau_C}}$,

$$\mathbb{E}\left\|\mathbf{x}^{(t)}-\mathbf{x}^{(t-\tau_t)}\right\|_2^2 \leq \frac{1}{4L^2\tau_C}\sum_{j=t-\tau_t}^{t-1}\mathbb{E}\left\|\nabla f(\mathbf{x}^{(j-\tau_j)})\right\|^2 + \tau_t\eta^2\sigma^2\,.$$

Summing over $T$,

$$\sum_{t=0}^{T}\mathbb{E}\left\|\mathbf{x}^{(t)}-\mathbf{x}^{(t-\tau_t)}\right\|_2^2 \leq \frac{1}{4L^2\tau_C}\sum_{t=0}^{T}\sum_{j=t-\tau_t}^{t-1}\mathbb{E}\left\|\nabla f(\mathbf{x}^{(j-\tau_j)})\right\|^2 + \sum_{t=0}^{T}\tau_t\eta^2\sigma^2$$

$$\leq \frac{1}{4L^2\tau_C}\sum_{t=0}^{T}\sum_{j=t-\tau_t}^{t-1}\mathbb{E}\left\|\nabla f(\mathbf{x}^{(j-\tau_j)})\right\|^2 + (T+1)\tau_{avg}\eta^2\sigma^2\,.$$

We now observe that the number of times each of the gradients $\left\|\nabla f(\mathbf{x}^{(j-\tau_j)})\right\|^2$ appears in the right hand side is bounded by $\tau_C^{(j)}-1$ because this many gradients started to be computed before the iteration $j$ and will get applied at some iteration $t > j$. Thus,

$$\sum_{t=0}^{T}\mathbb{E}\left\|\mathbf{x}^{(t)}-\mathbf{x}^{(t-\tau_t)}\right\|_2^2 \leq \frac{1}{4L^2}\sum_{t=0}^{T}\mathbb{E}\left\|\nabla f(\mathbf{x}^{(t-\tau_t)})\right\|^2 + (T+1)\frac{\sigma^2\eta}{L}\,,$$

where for the last $\sigma$ term we estimated $\eta \leq \frac{1}{2L\sqrt{\tau_{\max}\tau_C}}$ and used that both $\tau_{avg} \leq 2\tau_C$ and $\tau_{avg} \leq \tau_{\max}$. Dividing the inequality by $T+1$ we get the statement of the lemma. $\qquad\square$

Next, we give the proof of the first part of Theorem 6.

*Proof of Theorem 6, convergence rate* (8). We start by averaging with $T$ and dividing by $\eta$ the descent Lemma 13.

$$\frac{1}{T+1}\sum_{t=0}^{T}\left(\frac{1}{2}\mathbb{E}\left\|\nabla f(\mathbf{x}^{(t)})\right\|_2^2 + \frac{1}{4}\mathbb{E}\left\|\nabla f(\mathbf{x}^{(t-\tau_t)})\right\|^2\right) \leq \frac{1}{\eta(T+1)}\left(f(\mathbf{x}^{(0)})-f^\star\right) + L\eta\sigma^2$$

$$+ \frac{1}{T+1}\frac{L^2}{2}\sum_{t=0}^{T}\mathbb{E}\left\|\mathbf{x}^{(t)}-\mathbf{x}^{(t-\tau_t)}\right\|_2^2\,.$$

We next apply Lemma 14 to the last term and get

$$\frac{1}{T+1}\sum_{t=0}^{T}\left(\frac{1}{2}\mathbb{E}\left\|\nabla f(\mathbf{x}^{(t)})\right\|_2^2 + \frac{1}{4}\mathbb{E}\left\|\nabla f(\mathbf{x}^{(t-\tau_t)})\right\|^2\right) \leq \frac{1}{\eta(T+1)}\left(f(\mathbf{x}^{(0)})-f^\star\right) + L\eta\sigma^2$$

$$+ \frac{1}{8(T+1)}\sum_{t=0}^{T}\mathbb{E}\left\|\nabla f(\mathbf{x}^{(t-\tau_t)})\right\|^2 + \frac{L\eta\sigma^2}{2}\,.$$

And thus,

$$\frac{1}{T+1}\sum_{t=0}^{T}\mathbb{E}\left\|\nabla f(\mathbf{x}^{(t)})\right\|_2^2 \leq \frac{2}{\eta(T+1)}\left(f(\mathbf{x}^{(0)}) - f^\star\right) + 4L\eta\sigma^2.$$

It is only left to choose a stepsize $\eta$. Similar to previous works [48], we chose it as

$$\eta = \min\left\{\frac{1}{2L\sqrt{\tau_{\max}\tau_C}}; \left(\frac{r_0}{2L\sigma^2(T+1)}\right)^{\frac{1}{2}}\right\} \leq \frac{1}{2L\sqrt{\tau_{\max}\tau_C}},$$

where we defined $r_0 = f(\mathbf{x}^{(0)}) - f^\star$. With this choice of stepsize we indeed have that

- If $\frac{1}{2L\sqrt{\tau_{\max}\tau_C}} \leq \left(\frac{r_0}{2L\sigma^2(T+1)}\right)^{\frac{1}{2}}$ then $\eta = \frac{1}{2L\sqrt{\tau_{\max}\tau_C}}$, and

$$\frac{1}{T+1}\sum_{t=0}^{T}\mathbb{E}\left\|\nabla f(\mathbf{x}^{(t)})\right\|_2^2 \leq \frac{4Lr_0\sqrt{\tau_{\max}\tau_C}}{T+1} + \left(\frac{r_0}{2L\sigma^2(T+1)}\right)^{\frac{1}{2}}4L\sigma^2 = \mathcal{O}\left(\frac{\sigma}{\sqrt{T}} + \frac{\sqrt{\tau_{\max}\tau_C}}{T}\right)$$

- Otherwise if $\frac{1}{2L\sqrt{\tau_{\max}\tau_C}} > \left(\frac{r_0}{2L\sigma^2(T+1)}\right)^{\frac{1}{2}}$

$$\frac{1}{T+1}\sum_{t=0}^{T}\mathbb{E}\left\|\nabla f(\mathbf{x}^{(t)})\right\|_2^2 \leq 2\left(\frac{8L\sigma^2 r_0}{(T+1)}\right)^{\frac{1}{2}} = \mathcal{O}\left(\frac{\sigma}{\sqrt{T}}\right)$$

□

### A.3.2 Proof of Theorem 8

**Lemma 15** (Estimation of the residual). *Under Assumptions 1 and 3, the iterates of Algorithm 1 with the stepsizes $\eta_t$ chosen as in* (11)*, which we repeat here for readability*

$$\eta_t = \begin{cases} \eta & \tau_t \leq 2\tau_C, \\ < \min\{\eta, \frac{1}{4L\tau_t}\} & \tau_t > 2\tau_C, \end{cases}$$

*with $\eta \leq \frac{1}{4L\tau_C}$ satisfy*

$$\sum_{t=0}^{T}\eta_t\left\|\mathbf{x}^{(t)} - \mathbf{x}^{(t-\tau_t)}\right\|_2^2 \leq \frac{1}{16L^2}\sum_{t=0}^{T}\eta_t\left\|\nabla f(\mathbf{x}^{(t-\tau_t)})\right\|^2 + \frac{\sigma^2}{4L}\sum_{t=0}^{T}\eta_t^2.$$

*Proof.*

$$\eta_t\left\|\mathbf{x}^{(t)} - \mathbf{x}^{(t-\tau_t)}\right\|_2^2 = \eta_t\left\|\sum_{j=t-\tau_t}^{t-1}\eta_j\nabla F(\mathbf{x}^{(j-\tau_j)}, \xi_{j-\tau_j})\right\|^2 \overset{(2)}{\leq} \eta_t\left\|\sum_{j=t-\tau_t}^{t-1}\eta_j\nabla f(\mathbf{x}^{(j-\tau_j)})\right\|^2 + \eta_t\sum_{j=t-\tau_t}^{t-1}\eta_j^2\sigma^2$$

$$\overset{(18)}{\leq} \eta_t\tau_t\sum_{j=t-\tau_t}^{t-1}\eta_j^2\left\|\nabla f(\mathbf{x}^{(j-\tau_j)})\right\|^2 + \eta_t\sum_{j=t-\tau_t}^{t-1}\eta_j^2\sigma^2.$$

We use that each of the stepsizes $\eta_t \leq \frac{1}{4L\max\{\tau_t, \tau_C\}}$. Thus,

$$\eta_t\left\|\mathbf{x}^{(t)} - \mathbf{x}^{(t-\tau_t)}\right\|_2^2 \leq \frac{1}{4L}\sum_{j=t-\tau_t}^{t-1}\eta_j^2\left\|\nabla f(\mathbf{x}^{(j-\tau_j)})\right\|^2 + \frac{1}{4L\tau_C}\sum_{j=t-\tau_t}^{t-1}\eta_j^2\sigma^2.$$

Summing over $T$, and using that each of the gradients $\left\|\nabla f(\mathbf{x}^{j-\tau_j})\right\|^2$ would appear at most $\tau_C^{(j)} - 1$ times (see the discussion in the proof of Lemma 14)

$$\sum_{t=0}^{T}\eta_t\left\|\mathbf{x}^{(t)} - \mathbf{x}^{(t-\tau_t)}\right\|_2^2 \leq \frac{1}{4L}\sum_{t=0}^{T}\tau_C\eta_t^2\left\|\nabla f(\mathbf{x}^{(t-\tau_t)})\right\|^2 + \frac{\sigma^2}{4L}\sum_{t=0}^{T}\eta_t^2.$$

Using again that $\eta_t \leq \frac{1}{4L\max\{\tau_t, \tau_C\}}$ we get the statement of the lemma. □

*Proof of Theorem 8.* We start by summing the descent Lemma 13 over the iterations $t = 0, \ldots, T$.

$$\sum_{t=0}^{T} \eta_t \left( \frac{1}{2} \mathbb{E} \left\| \nabla f(\mathbf{x}^{(t)}) \right\|_2^2 + \frac{1}{4} \mathbb{E} \left\| \nabla f(\mathbf{x}^{(t-\tau_t)}) \right\|^2 \right) \leq \left( f(\mathbf{x}^{(0)}) - f^\star \right) + L\sigma^2 \sum_{t=0}^{T} \eta_t^2 + \frac{L^2}{2} \sum_{t=0}^{T} \eta_t \left\| \mathbf{x}^{(t)} - \mathbf{x}^{(t-\tau_t)} \right\|_2^2 .$$

Next, we substitute Lemma 15 into the last term,

$$\sum_{t=0}^{T} \eta_t \left( \frac{1}{2} \mathbb{E} \left\| \nabla f(\mathbf{x}^{(t)}) \right\|_2^2 + \frac{1}{4} \mathbb{E} \left\| \nabla f(\mathbf{x}^{(t-\tau_t)}) \right\|^2 \right) \leq \left( f(\mathbf{x}^{(0)}) - f^\star \right) + L\sigma^2 \sum_{t=0}^{T} \eta_t^2$$

$$+ \frac{1}{32} \sum_{t=0}^{T} \eta_t \left\| \nabla f(\mathbf{x}^{(t-\tau_t)}) \right\|^2 + \frac{\sigma^2 L}{8} \sum_{t=0}^{T} \eta_t^2 .$$

Rearranging we thus get

$$\sum_{t=0}^{T} \eta_t \mathbb{E} \left\| \nabla f(\mathbf{x}^{(t)}) \right\|_2^2 \leq 2 \left( f(\mathbf{x}^{(0)}) - f^\star \right) + 4L\sigma^2 \sum_{t=0}^{T} \eta_t^2 .$$

We note that due to our choice of stepsizes (11), $\eta_t \leq \eta$, it also holds that $\sum_{t=0}^{T} \eta_t \geq \sum_{t:\tau_t \leq 2\tau_C} \eta \geq \frac{T+1}{2}\eta$ since there are at least half of the iterations with the delay smaller than two times the average. Using this, we estimate

$$\frac{1}{\sum_{t=0}^{T} \eta_t} \sum_{t=0}^{T} \eta_t \mathbb{E} \left\| \nabla f(\mathbf{x}^{(t)}) \right\|_2^2 \leq \frac{4}{(T+1)\eta} \left( f(\mathbf{x}^{(0)}) - f^\star \right) + 8L\sigma^2\eta^2 .$$

It remains to tune the stepsize $\eta$, i.e. to pick is such as to minimize the right hand side of this expression. See Lemma 17 in [26]. $\qquad \square$

## A.4 Proof of Theorem 6, convergence rate (9)

To prove the last claim of Theorem 6 we take another approach and follow the perturbed iterate analysis [31].

We introduce a virtual sequence $\tilde{\mathbf{x}}^t$ defined as

$$\tilde{\mathbf{x}}^{(0)} = \mathbf{x}^{(0)}, \qquad\qquad \tilde{\mathbf{x}}^{(t+1)} = \tilde{\mathbf{x}}^{(t)} - \eta \sum_{i \in \mathcal{A}_t} \nabla F(\mathbf{x}^{(t)}, \xi_t),$$

where we define $\mathcal{A}_0 := \mathcal{C}_0$, and $\hat{\tau}_t^i$ is the delay with which the corresponding gradient will be computed. That is, if we denote $j = t + \hat{\tau}_i^t$, then it will hold that $j - \tau_j = t$. This defines a virtual sequence and we do not have access to it during the execution of Algorithm 1.

**Lemma 16** (Descent lemma). *Under Assumptions 1 and 3, if in Algorithm 1 the stepsize $\eta_t < \frac{1}{2L\tau_C}$ then it holds that*

$$\mathbb{E}_{t+1} f(\tilde{\mathbf{x}}^{(t+1)}) \leq f(\tilde{\mathbf{x}}^{(t)}) - \frac{\eta}{4}|\mathcal{A}_t| \left\| \nabla f(\mathbf{x}^{(t)}) \right\|_2^2 + \frac{\eta}{2}|\mathcal{A}_t|L^2 \left\| \mathbf{x}^{(t)} - \tilde{\mathbf{x}}^{(t)} \right\|^2 + \frac{L\eta^2\sigma^2|\mathcal{A}_t|}{2} .$$

*Proof.* Because function $f$ is $L$-smooth, we have

$$\mathbb{E}_{t+1} f(\tilde{\mathbf{x}}^{(t+1)}) = \mathbb{E}_{t+1} f\left( \tilde{\mathbf{x}}^{(t)} - \eta \sum_{i \in \mathcal{A}_t} \nabla F(\mathbf{x}^{(t)}, \xi^{(t)}) \right)$$

$$\leq f(\tilde{\mathbf{x}}^{(t)}) - \eta|\mathcal{A}_t| \underbrace{\langle \nabla f(\tilde{\mathbf{x}}^{(t)}), \nabla f(\mathbf{x}^{(t)}) \rangle}_{=:T_1} + \mathbb{E}_{t+1} \frac{L}{2}\eta^2 \underbrace{\left\| \sum_{i \in \mathcal{A}_t} \nabla F(\mathbf{x}^{(t)}, \xi_t) \right\|_2^2}_{=:T_2} .$$

We estimate the second term as

$$T_1 = -\left\langle \nabla f(\mathbf{x}^{(t)}), \nabla f(\tilde{\mathbf{x}}^{(t)}) \right\rangle = -\frac{1}{2} \left\| \nabla f(\mathbf{x}^{(t)}) \right\|^2 - \frac{1}{2} \left\| \nabla f(\tilde{\mathbf{x}}^{(t)}) \right\|^2 + \frac{1}{2} \left\| \nabla f(\mathbf{x}^{(t)}) - \nabla f(\tilde{\mathbf{x}}^{(t)}) \right\|^2$$

$$\leq -\frac{1}{2} \left\| \nabla f(\mathbf{x}^{(t)}) \right\|^2 + \frac{1}{2} \left\| \nabla f(\mathbf{x}^{(t)}) - \nabla f(\tilde{\mathbf{x}}^{(t)}) \right\|^2 .$$

For the last term, using the notation $\pm a = a - a = 0 \ \forall a$,

$$T_2 = \mathbb{E}_{t+1} \left\| \sum_{i \in \mathcal{A}_t} \nabla F(\mathbf{x}^{(t)}, \xi_t) \pm |\mathcal{A}_t| \nabla f(\mathbf{x}^{(t)}) \right\|_2^2$$

$$\overset{(2)}{\leq} |\mathcal{A}_t| \sigma^2 + |\mathcal{A}_t|^2 \left\| \nabla f(\mathbf{x}^{(t)}) \right\|_2^2 .$$

Combining this together, using $L$-smoothness to estimate $\left\| \nabla f(\mathbf{x}^{(t)}) - \nabla f(\tilde{\mathbf{x}}^{(t)}) \right\|_2^2$ we get

$$\mathbb{E}_{t+1} \ f(\tilde{\mathbf{x}}^{(t+1)}) \leq f(\tilde{\mathbf{x}}^{(t)}) - \left( \frac{\eta}{2} |\mathcal{A}_t| - \frac{\eta^2 L |\mathcal{A}_t|^2}{2} \right) \left\| \nabla f(\mathbf{x}^{(t)}) \right\|_2^2 + \frac{\eta}{2} |\mathcal{A}_t| L^2 \left\| \mathbf{x}^{(t)} - \tilde{\mathbf{x}}^{(t)} \right\|^2 + \frac{L \eta^2 \sigma^2 |\mathcal{A}_t|}{2} .$$

Using that $\eta \leq \frac{1}{2L\tau_C} \leq \frac{1}{2L|\mathcal{A}_t|}$ we get statement of the Lemma. $\qquad \square$

**Lemma 17** (Estimation of the residual). *Under Assumptions 1, 3, iterated of Algorithm 1 with the constant stepsize $\eta_t \equiv \eta$ with $\eta \leq \frac{1}{2L\tau_C}$ satisfy*

$$\mathbb{E} \left\| \mathbf{x}^{(t)} - \tilde{\mathbf{x}}^{(t)} \right\|_2^2 \leq \tau_C^2 \eta^2 G^2 + \eta^2 \tau_C \sigma^2 .$$

*Proof.*

$$\mathbb{E} \left\| \mathbf{x}^{(t)} - \tilde{\mathbf{x}}^{(t)} \right\|_2^2 = \mathbb{E} \left\| \sum_{j \in \mathcal{C}_t} \eta \nabla F(\mathbf{x}^{(j)}, \xi_j) \right\|_2^2 \overset{(2)}{\leq} \mathbb{E} \left\| \sum_{j \in \mathcal{C}_t} \eta \nabla f(\mathbf{x}^{(j)}) \right\|_2^2 + \eta^2 \tau_C^{(t)} \sigma^2$$

$$\overset{(18)}{\leq} \tau_C^{(t)} \sum_{j \in \mathcal{C}_t} \eta^2 \, \mathbb{E} \left\| \nabla f(\mathbf{x}^{(j)}) \right\|_2^2 + \eta^2 \tau_C^{(t)} \sigma^2$$

$$\overset{(5)}{\leq} (\tau_C^{(t)})^2 \eta^2 G^2 + \eta^2 \tau_C^{(t)} \sigma^2 . \qquad \square$$

We are now ready to prove the second claim of Theorem 6.

*Proof of Theorem 6, convergence rate* (9). We start by summing over $t = 0, \dots, T$ the descent Lemma 16. We also divide it by $\eta$,

$$\sum_{t=0}^{T} \frac{1}{4} |\mathcal{A}_t| \left\| \nabla f(\mathbf{x}^{(t)}) \right\|_2^2 \leq \frac{1}{\eta} \left( f(\mathbf{x}^{(0)}) - f^\star \right) + \frac{L\eta\sigma^2}{2} \sum_{t=0}^{T} |\mathcal{A}_t| + \frac{L^2}{2} \sum_{t=0}^{T} |\mathcal{A}_t| \left\| \mathbf{x}^{(t)} - \tilde{\mathbf{x}}^{(t)} \right\|^2 .$$

We further use Lemma 17 for the last term

$$\sum_{t=0}^{T} \frac{1}{4} |\mathcal{A}_t| \left\| \nabla f(\mathbf{x}^{(t)}) \right\|_2^2 \leq \frac{1}{\eta} \left( f(\mathbf{x}^{(0)}) - f^\star \right) + \frac{L\eta\sigma^2}{2} \sum_{t=0}^{T} |\mathcal{A}_t| + \frac{L^2}{2} \left( \tau_C^2 \eta^2 G^2 + \eta^2 \tau_C \sigma^2 \right) \sum_{t=0}^{T} |\mathcal{A}_t| .$$

We further use that $\eta \leq \frac{1}{2L\tau_C}$ for the last $\sigma$ term and divide the full inequality by $\frac{1}{4} \mathcal{W}_T$, where we defined $\mathcal{W}_T = \sum_{t=0}^{T} |\mathcal{A}_t|$

$$\frac{1}{\mathcal{W}_T} \sum_{t=0}^{T} |\mathcal{A}_t| \left\| \nabla f(\mathbf{x}^{(t)}) \right\|_2^2 \leq \frac{4}{\eta \mathcal{W}_T} \left( f(\mathbf{x}^{(0)}) - f^\star \right) + 4L\eta\sigma^2 + 2L^2 \tau_C^2 \eta^2 G^2 .$$

Note that because at every step $t$ only one of the gradients is getting applied, $T \leq \sum_{t=0}^{T} |\mathcal{A}_t| \leq T + \tau_C \leq 2T$ for $T \geq \tau_C$.

It is left to tune the stepsize using Lemma 17 in [26] to get the final convergence rate. $\qquad \square$

## A.5 Proof of the Theorem 11

We first re-state the theorem

**Theorem 11** (constant stepsizes). *Under Assumptions 1, 2, 3, 5 there exists a constant stepsize $\eta_t \equiv \eta$ such that for Algorithm 2 it holds that $\frac{1}{T+1}\sum_{t=0}^{T}\left\|\nabla f(\mathbf{x}^{(t)})\right\|_2^2 \leq \varepsilon$ after*

$$\mathcal{O}\left(\frac{\sigma^2}{\varepsilon^2} + \frac{\zeta^2}{\varepsilon^2} + \frac{\sqrt{\tau_{avg}\frac{1}{n}\sum_{i=1}^{n}\zeta_i^2\tau_{avg}^i}}{\varepsilon^{\frac{3}{2}}} + \frac{\sqrt{\tau_{avg}\tau_{\max}}}{\varepsilon}\right) \qquad \text{iterations,} \qquad (14)$$

*Under Assumptions 1, 2, 3 and additional bounded gradient Assumption 4, it holds that $\frac{1}{T+1}\sum_{t=0}^{T}\left\|\nabla f(\mathbf{x}^{(t)})\right\|_2^2 \leq \varepsilon$ after*

$$\mathcal{O}\left(\frac{\sigma^2}{\varepsilon^2} + \frac{\zeta^2}{\varepsilon^2} + \frac{\tau_{avg}G}{\varepsilon^{\frac{3}{2}}} + \frac{\tau_{avg}}{\varepsilon}\right) \qquad \text{iterations.} \qquad (15)$$

We utilize again the perturbed iterate technique [31]. We introduce a virtual sequence $\tilde{\mathbf{x}}^{(t)}$ as

$$\tilde{\mathbf{x}}^{(0)} = \mathbf{x}^{(0)} \qquad\qquad \tilde{\mathbf{x}}^{(t+1)} = \tilde{\mathbf{x}}^{(t)} - \eta\nabla F_{k_t}(\mathbf{x}^{(t)}, \xi_t),$$

where we define $\hat{\tau}_t$ as the delay with which the corresponding gradient will be computed. If we denote $j = t + \hat{\tau}_t$, then it holds that $j - \tau_j = t$.

**Lemma 18** (Descent Lemma). *Under Assumptions 1, 2, 3, for Algorithm 2 with the stepsize $\eta_t \leq \frac{1}{4L}$ it holds that*

$$\mathbb{E}_{t+1}\, f(\tilde{\mathbf{x}}^{(t+1)}) \leq f(\tilde{\mathbf{x}}^{(t)}) - \frac{\eta}{4}\left\|\nabla f(\mathbf{x}^{(t)})\right\|_2^2 + \frac{L\eta^2\sigma^2}{2} + L\eta^2\zeta^2 + \frac{\eta L^2}{2}\left\|\mathbf{x}^{(t)} - \tilde{\mathbf{x}}^{(t)}\right\|_2^2. \quad (19)$$

*Proof.* Because the function $f$ is $L$-smooth, we have

$$\mathbb{E}_{t+1}\, f(\tilde{\mathbf{x}}^{(t+1)}) = \mathbb{E}_{t+1}\, f\left(\tilde{\mathbf{x}}^{(t)} - \eta\nabla F_{k_t}(\mathbf{x}^{(t)}, \xi_t)\right)$$

$$\leq f(\tilde{\mathbf{x}}^{(t)}) - \eta\underbrace{\langle\nabla f(\tilde{\mathbf{x}}^{(t)}), \nabla f(\mathbf{x}^{(t)})\rangle}_{=:T_1} + \mathbb{E}_{t+1}\,\frac{L}{2}\eta^2\underbrace{\left\|\nabla F_{k_t}(\mathbf{x}^{(t)}, \xi_t)\right\|_2^2}_{=:T_2},$$

where expectation is taken over both the stochastic noise $\xi$ and sampled index $j_t$. We estimate terms $T_1$ and $T_2$ separately

$$T_1 = -\frac{\eta}{2}\left\|\nabla f(\mathbf{x}^{(t)})\right\|^2 - \frac{\eta}{2}\left\|\nabla f(\tilde{\mathbf{x}}^{(t)})\right\|^2 + \frac{\eta}{2}\left\|\nabla f(\mathbf{x}^{(t)}) - \nabla f(\tilde{\mathbf{x}}^{(t)})\right\|^2$$

$$\leq -\frac{\eta}{2}\left\|\nabla f(\mathbf{x}^{(t)})\right\|^2 + \frac{\eta}{2}\left\|\nabla f(\mathbf{x}^{(t)}) - \nabla f(\tilde{\mathbf{x}}^{(t)})\right\|^2.$$

For the last term, using the notation $\pm a = a - a = 0\ \forall a$,

$$T_2 = \mathbb{E}_{t+1}\left\|\nabla F_{k_t}(\mathbf{x}^{(t)}, \xi_t) \pm \nabla f_{j_t}(\mathbf{x}^{(t)}) \pm \nabla f(\mathbf{x}^{(t)})\right\|_2^2$$

$$\overset{(2)}{\leq} \sigma^2 + 2\mathbb{E}_{k_t}\left\|\nabla f_{k_t}(\mathbf{x}^{(t)}) - \nabla f(\mathbf{x}^{(t)})\right\|_2^2 + 2\left\|\nabla f(\mathbf{x}^{(t)})\right\|_2^2$$

$$\overset{(3)}{\leq} \sigma^2 + 2\zeta^2 + 2\left\|\nabla f(\mathbf{x}^{(t)})\right\|_2^2.$$

Combining this together and using $L$-smoothness to estimate $\left\|\nabla f(\mathbf{x}^{(t)}) - \nabla f(\tilde{\mathbf{x}}^{(t)})\right\|_2^2$ we get

$$\mathbb{E}_{t+1}\, f(\tilde{\mathbf{x}}^{(t+1)}) \leq f(\tilde{\mathbf{x}}^{(t)}) - \left(\frac{\eta}{2} - L\eta^2\right)\left\|\nabla f(\mathbf{x}^{(t)})\right\|_2^2 + \frac{\eta}{2}L^2\left\|\mathbf{x}^{(t)} - \tilde{\mathbf{x}}^{(t)}\right\|^2 + \frac{L\eta^2\sigma^2}{2} + L\eta^2\zeta^2.$$

Applying $\eta \leq \frac{1}{4L}$ we get statement of the lemma. $\qquad\square$

### A.5.1 Proof of Theorem 11, convergence rate (14)

**Lemma 19** (Estimation of the distance $\left\|\mathbf{x}^{(t)} - \tilde{\mathbf{x}}^{(t)}\right\|_2^2$). *Under Assumptions 1, 2, 3, for Algorithm 2 with the stepsize $\eta_t \leq \frac{1}{4L\sqrt{\tau_C \tau_{\max}}}$ it holds that*

$$\frac{1}{T+1}\sum_{t=0}^{T} \mathbb{E}\left\|\mathbf{x}^{(t)} - \tilde{\mathbf{x}}^{(t)}\right\|_2^2 \leq \frac{\eta\sigma^2}{4L} + \frac{2\eta^2\tau_C}{T+1}\frac{1}{n}\sum_{j=1}^{n}\zeta_j^2\bar{\tau}_j + \frac{1}{8L^2(T+1)}\sum_{t=0}^{T}\mathbb{E}\left\|\nabla f(\mathbf{x}^{(t)})\right\|_2^2.$$

*Proof.*

$$\mathbb{E}\left\|\mathbf{x}^{(t)} - \tilde{\mathbf{x}}^{(t)}\right\|_2^2 = \mathbb{E}\,\eta^2 \left\|\sum_{i\in\mathcal{C}_t}\nabla F_{j_i}(\mathbf{x}^{(i)},\xi_i)\right\|_2^2 \stackrel{(2)}{\leq} \eta^2\tau_C\sigma^2 + \eta^2\,\mathbb{E}\left\|\sum_{i\in\mathcal{C}_t}\nabla f_{j_i}(\mathbf{x}^{(i)})\right\|_2^2$$

$$\stackrel{(18)}{\leq} \eta^2\tau_C\sigma^2 + 2\eta^2\,\mathbb{E}\left\|\sum_{i\in\mathcal{C}_t}\nabla f_{j_i}(\mathbf{x}^{(i)}) - \nabla f(\mathbf{x}^{(i)})\right\|_2^2 + 2\eta^2\,\mathbb{E}\left\|\sum_{i\in\mathcal{C}_t}\nabla f(\mathbf{x}^{(i)})\right\|_2^2$$

$$\stackrel{(18)}{\leq} \eta^2\tau_C\sigma^2 + 2\eta^2\tau_C^{(t)}\,\mathbb{E}\sum_{i\in\mathcal{C}_t}\zeta_{j_i}^2 + 2\eta^2\tau_C\,\mathbb{E}\sum_{i\in\mathcal{C}_t}\left\|\nabla f(\mathbf{x}^{(i)})\right\|_2^2.$$

Averaging over $T$, we get

$$\frac{1}{T+1}\sum_{t=0}^{T}\mathbb{E}\left\|\mathbf{x}^{(t)} - \tilde{\mathbf{x}}^{(t)}\right\|_2^2 \leq \eta^2\tau_C\sigma^2 + 2\eta^2\tau_C\frac{1}{T+1}\sum_{t=0}^{T}\mathbb{E}\sum_{i\in\mathcal{C}_t}\zeta_{j_i}^2 + 2\eta^2\tau_C\frac{1}{T+1}\sum_{t=0}^{T}\mathbb{E}\sum_{i\in\mathcal{C}_t}\left\|\nabla f(\mathbf{x}^{(i)})\right\|_2^2.$$

We note that in the second term each of $\zeta_j$ appears exactly $\tau_j^{sum}$ times, where $\tau_j^{sum}$ is the sum of the all the delays that happened on the node $j$. In the last term, we estimate the number of appearance of each of $\left\|\nabla f(\mathbf{x}^{(i)})\right\|_2^2$ bt $\tau_{\max}$, thus

$$\frac{1}{T+1}\sum_{t=0}^{T}\mathbb{E}\left\|\mathbf{x}^{(t)} - \tilde{\mathbf{x}}^{(t)}\right\|_2^2 \leq \eta^2\tau_C\sigma^2 + 2\eta^2\tau_C\,\mathbb{E}\,\frac{1}{T+1}\sum_{j=1}^{n}\zeta_j^2\tau_j^{sum} + 2\eta^2\tau_C\tau_{\max}\frac{1}{T+1}\sum_{t=0}^{T}\mathbb{E}\left\|\nabla f(\mathbf{x}^{(t)})\right\|_2^2,$$

we further use that number of times $T_j$ that every node $j$ got sampled are equal in expectation because of uniform sampling in line 6 of Algorithm 2. Thus,

$$\frac{1}{T+1}\sum_{t=0}^{T}\mathbb{E}\left\|\mathbf{x}^{(t)} - \tilde{\mathbf{x}}^{(t)}\right\|_2^2 \leq \eta^2\tau_C\sigma^2 + 2\eta^2\tau_C\frac{1}{T+1}\frac{1}{n}\sum_{j=1}^{n}\zeta_j^2\bar{\tau}_j + 2\eta^2\tau_C\tau_{\max}\frac{1}{T+1}\sum_{t=0}^{T}\mathbb{E}\left\|\nabla f(\mathbf{x}^{(t)})\right\|_2^2.$$

Using that $\eta \leq \frac{1}{4L\sqrt{\tau_C\tau_{\max}}}$ we get the statement of the lemma. $\qquad\square$

*Proof of Theorem 11,* (14). First, averaging the descent Lemma 16,

$$\frac{1}{T+1}\sum_{t=0}^{T}\mathbb{E}\left\|\nabla f(\mathbf{x}^{(t)})\right\|_2^2 \leq \frac{4}{\eta(T+1)}\left(f(\mathbf{x}^0) - f(\mathbf{x}^T)\right) + 2L\eta\sigma^2 + 4L\eta\zeta^2 + \frac{2L^2}{T+1}\sum_{t=0}^{T}\mathbb{E}\left\|\mathbf{x}^{(t)} - \tilde{\mathbf{x}}^{(t)}\right\|_2^2.$$

Now plugging in the result of Lemma 19, we get

$$\frac{1}{T+1}\sum_{t=0}^{T}\mathbb{E}\left\|\nabla f(\mathbf{x}^{(t)})\right\|_2^2 \leq \frac{4}{\eta(T+1)}\left(f(\mathbf{x}^0) - f(\mathbf{x}^T)\right) + 2L\eta\sigma^2 + 4L\eta\zeta^2 + \frac{L\eta\sigma^2}{2}$$

$$+ \frac{4L^2\eta^2\tau_C}{T+1}\frac{1}{n}\sum_{j=1}^{n}\zeta_j^2\bar{\tau}_j + \frac{1}{4(T+1)}\sum_{t=0}^{T}\mathbb{E}\left\|\nabla f(\mathbf{x}^{(t)})\right\|_2^2.$$

Rearranging terms we thus get

$$\frac{1}{2(T+1)}\sum_{t=0}^{T}\mathbb{E}\left\|\nabla f(\mathbf{x}^{(t)})\right\|_2^2 \leq \frac{4}{\eta(T+1)}\left(f(\mathbf{x}^0) - f(\mathbf{x}^T)\right) + 3L\eta\sigma^2 + 4L\eta\zeta^2 + \frac{4L^2\eta^2\tau_C}{T+1}\frac{1}{n}\sum_{j=1}^{n}\zeta_j^2\bar{\tau}_j$$

It is only left to tune the stepsize using Lemma 17 in [26]. $\qquad\square$

### A.5.2 Proof of Theorem 11, convergence rate (15).

**Lemma 20** (Estimation of the distance $\left\| \mathbf{x}^{(t)} - \tilde{\mathbf{x}}^{(t)} \right\|_2^2$). *Under Assumptions 1, 2, 3, 4 for Algorithm 2 with the stepsize $\eta_t \equiv \eta \leq \frac{1}{4L\tau_C}$ it holds that*

$$\frac{1}{T+1} \sum_{t=0}^{T} \mathbb{E} \left\| \mathbf{x}^{(t)} - \tilde{\mathbf{x}}^{(t)} \right\|_2^2 \leq \frac{\eta\sigma^2}{4L} + \eta^2 \tau_C^2 G^2 \,.$$

*Proof.* We start our proof similar way as before

$$
\begin{aligned}
\mathbb{E} \left\| \mathbf{x}^{(t)} - \tilde{\mathbf{x}}^{(t)} \right\|_2^2 = \mathbb{E}\, \eta^2 \left\| \sum_{i \in \mathcal{C}_t} \nabla F_{j_i}(\mathbf{x}^{(i)}, \xi_i) \right\|_2^2 &\overset{(2)}{\leq} \eta^2 \tau_C \sigma^2 + \eta^2 \, \mathbb{E} \left\| \sum_{i \in \mathcal{C}_t} \nabla f_{j_i}(\mathbf{x}^{(i)}) \right\|_2^2 \\
&\overset{(18)}{\leq} \eta^2 \tau_C \sigma^2 + \eta^2 \tau_C \sum_{i \in \mathcal{C}_t} \mathbb{E} \left\| \nabla f_{j_i}(\mathbf{x}^{(i)}) \right\|_2^2 \\
&\overset{(5)}{\leq} \eta^2 \tau_C \sigma^2 + \eta^2 \tau_C^2 G^2 \\
&\leq \frac{\eta\sigma^2}{4L} + \eta^2 \tau_C^2 G^2
\end{aligned}
$$

where on the last line we used that stepsize $\eta \leq \frac{1}{4L\tau_C}$. $\qquad\square$

*Proof of the Theorem 11, (15).* We start by averaging the descent Lemma 16,

$$\frac{1}{T+1} \sum_{t=0}^{T} \mathbb{E} \left\| \nabla f(\mathbf{x}^{(t)}) \right\|_2^2 \leq \frac{4}{\eta(T+1)} \left( f(\mathbf{x}^0) - f(\mathbf{x}^T) \right) + 2L\eta\sigma^2 + 4L\eta\zeta^2 + \frac{2L^2}{T+1} \sum_{t=0}^{T} \mathbb{E} \left\| \mathbf{x}^{(t)} - \tilde{\mathbf{x}}^{(t)} \right\|_2^2 \,.$$

We now plug in the results of Lemma 20 and get

$$\frac{1}{T+1} \sum_{t=0}^{T} \mathbb{E} \left\| \nabla f(\mathbf{x}^{(t)}) \right\|_2^2 \leq \frac{4}{\eta(T+1)} \left( f(\mathbf{x}^0) - f(\mathbf{x}^T) \right) + 3L\eta\sigma^2 + 4L\eta\zeta^2 + 2L^2 \eta^2 \tau_C^2 G^2 \,.$$

It is only left to tune the stepsize using Lemma 17 in [26]. $\qquad\square$

## B   Estimating Speedup over Synchronous SGD

Assume that we have $n$ workers with identical functions $f_i \equiv f_j \equiv f \ \forall i, j$ (homogeneous case). Assume that the objective function has a sum-structure $f(\mathbf{x}) = \frac{1}{m} \sum_{j=1}^{m} F(\mathbf{x}, \xi_j)$. This setting is common in machine learning where each $F(\mathbf{x}, \xi_j)$ represent a loss function of a model $\mathbf{x}$ on a datapoint $\xi_j \in \mathcal{D}, |\mathcal{D}| = m$. Assume that to compute stochastic gradient $\nabla F(\mathbf{x}, \xi_j)$ each worker needs a constant time $\Delta_j$. W.l.o.g. we assume that $\Delta_i$ are ordered as $\Delta_1 \leq \Delta_2 \leq \cdots \leq \Delta_m$.

**Lemma 21.** *In expectation, the asynchronous Algorithm 2 needs*

$$\bar{\Delta} = \frac{1}{m} \sum_{i=1}^{m} \Delta_i$$

*time to compute $n$ gradients, while mini-batch SGD with batch size $n$ needs*

$$\tilde{\Delta} = \sum_{i=1}^{m} \alpha_i \Delta_i$$

*time to compute a batch of $n$ gradients, where $\alpha_i = \frac{i^n - (i-1)^n}{m^n}$. It is also always holds that $\bar{\Delta} \leq \tilde{\Delta}$.*

With this lemma we can precisely estimate how much faster the asynchronous algorithm is compared to the classic synchronous mini-batch one. Note that $\alpha_i$ are increasing with $i$ with a rate of $\mathcal{O}(i^n)$, thus in mini-batch SGD, the large delays get a much higher weight than the small delays, especially when the batch size $n$ is large.

For example, consider 1000 clients, 900 of which compute their update every $10s$, while 100 of them computes their update every $60s$. Then the expected time for $n$ gradients of the asynchronous algorithm will be 15s, while synchronous mini-batch SGD (with $n = 10$) will take a significantly longer time of 42.5s for the same number of gradients.

## B.1 Proof of Lemma 21

In this section we prove Lemma 21.

*Proof.* We start by proving the first claim.

**Time of Asynchronous Algorithm 2.** Assume the concurrency is $n = 1$. Then, Algorithm 2 is synchronous, and as we sample every client with equal probability (line 6 of Algorithm 2), the expected time to compute one gradient is equal to $\frac{1}{m} \sum_{i=1}^{m} \Delta_i$.

To calculate the estimated time with concurrency $n > 1$ we can view the Algorithm 2 as having $n$ independent copies of the previous process run in parallel. Thus, in the same time $\frac{1}{m} \sum_{i=1}^{m} \Delta_i$ in expectation Algorithm 2 will compute $n$ gradients.

**Time of Mini-batch SGD.** The expected time of mini-batch SGD of size $n$ is equal to $\mathbb{E} \max\{\Delta_{i_1}, \dots, \Delta_{i_n}\}$ with each $i_j \sim \text{Uniform}[1, m]$. Denote a random variable $X = \max\{\Delta_{i_1}, \dots, \Delta_{i_n}\}$ that takes values within $\Delta_1, \dots \Delta_m$. Since $i_j$ are independent from each other,

$$\Pr[X \leq \Delta_k] = \prod_{j=1}^{n} \Pr[\Delta_{i_j} \leq \Delta_k] = \prod_{j=1}^{n} \Pr[X \leq \Delta_k] = \prod_{j=1}^{n} \Pr[i_j \leq k] = \left(\frac{k}{m}\right)^n.$$

Thus,

$$\Pr[X = \Delta_k] = \frac{k^n - (k-1)^n}{m^n}.$$

And therefore,

$$\mathbb{E}\,X = \sum_{k=1}^{m} \Pr[X = \Delta_k]\,\Delta_k. \qquad \square$$

## C Experiments

In this set of experiments we aim to illustrate the dependence on the maximum delay $\tau_{\max}$ in Theorem 6, as depicted in Equation (8). For this, we set the stochastic noise $\sigma$ to zero. In this case Theorem 6, Equation (8) predicts that to reach an $\varepsilon$ accuracy, Algorithm 1 needs $T = \mathcal{O}\left(\frac{\sqrt{\tau_{\max}\tau_C}}{\varepsilon}\right)$ iterations. In our experiments we fix $\tau_C = 2$, $\varepsilon = 10^{-14}$. Since $\tau_C = 2$, we have two workers. We vary the relative speed of the second worker, and thus affecting the maximum delay: if the second worker is $x$ times slower than the first worker, then the maximum delay $\tau_{\max} = x$. We measure the time $T$ to reach the accuracy $\varepsilon$. Since all the other parameters are constant, it holds that $T = C_1\sqrt{\tau_{\max}}$.

We perform experiments on two different functions:

(i) quadratic function $f(\mathbf{x}) = \frac{1}{2}\|A\mathbf{x} - \mathbf{b}\|_2^2$, $\mathbf{x}, \mathbf{b} \in \mathbb{R}^{10}$, $b_i \sim \mathcal{N}(0, 1)$, $i \in [1, 10]$, $A \in \mathbb{R}^{10 \times 10}$ is a random matrix with $\lambda_{\max}(A) = 2$, $\lambda_{\min}(A) = 1$ and the rest of eigenvalues are equally spaced in between.

(ii) logistic regression function $f(\mathbf{x}) = \frac{1}{m} \sum_{j=1}^{m} \log(1 + \exp(-b_j \mathbf{a}_j^\top \mathbf{x}))$, where each $b_j$ is sampled uniformly at random from the set $\{-1, 1\}$, and $\mathbf{a}_j \sim \mathcal{N}(0, 1)^{20}$, $\mathbf{x} \in \mathbb{R}^{20}$, $m = 100$.

We estimate the error as the average over the last 30 iterations $\hat{\varepsilon}_T = \frac{1}{30} \sum_{i=0}^{29} \|\nabla f(\mathbf{x}_{T-i})\|_2$. We tune the stepsize $\eta$ for every experiment *separately* over the logarithmic grid between $10^{-5}$ and $10^2$ ensuring that the optimal stepsize value is not on the edge of the grid.

Figure 2 shows the resulting dependence of $T$ on $\tau_{\max}$ for the quadratic function (i), and Figure 3 for the logistic regression function (ii). In both cases we see that $T$ has linear dependence on $\sqrt{\tau_{\max}}$ confirming our theory.

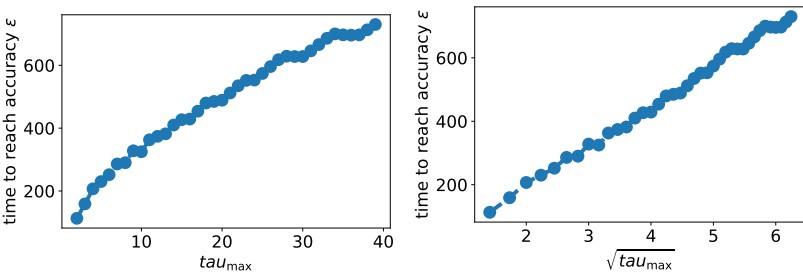

Figure 2: Verification of $\sqrt{\tau_{\max}}$ dependence on random quadratic function (i). We see that $T$ has linear dependence on $\sqrt{\tau_{\max}}$

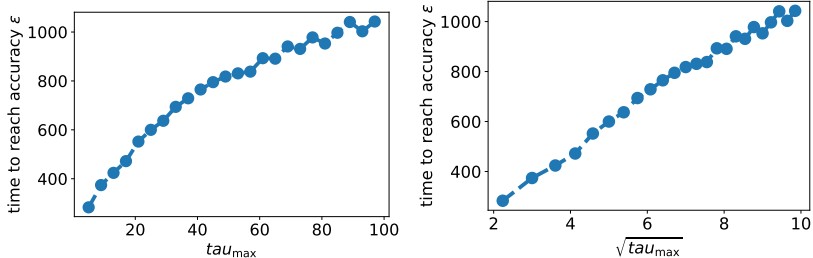

Figure 3: Verification of $\sqrt{\tau_{\max}}$ dependence on random logistic regression function (ii). We see that $T$ has linear dependence on $\sqrt{\tau_{\max}}$