# OpenReview forum: "Sharper Convergence Guarantees for Asynchronous SGD for Distributed and Federated Learning"
_NeurIPS.cc/2022/Conference — NeurIPS 2022 Accept_

### Official Review · Reviewer_88dr · 2022-06-26

**Rating:** 8
**Confidence:** 4
**Soundness:** 4 excellent
**Presentation:** 4 excellent
**Contribution:** 4 excellent

**Summary:**

The authors provide the analysis of Asynchronous SGD algorithms in the homogeneous and heterogeneous regimes. First, in the homogeneous regime, they provide a better analysis of Asynchronous SGD and get better rates without additional changes in the original algorithm with a constant step size. Next, they omit the dependence on the maximum delay by exploiting the delay-adaptive learning rate scheme. Finally, the authors provide an analysis of the heterogeneous regime, where the dependence on the maximum delay is improved, or the dependence can be omitted under the gradient boundness assumption.

**Questions:**

Minor problems:
1. It is better to check terms $\xi_t$ and $\xi_{t - \tau}.$ For instance, in Algorithm 1, I would expect to see $\nabla F(x^{t - \tau_t}, \xi_{t - \tau_t})$ because $\xi_{t - \tau_t}$ was generated at time $t - \tau_t.$
2. 640: The line says $\tau_{avg} \leq \tau_C.$ Does it hold for every $T$? Maybe, as in Remark 5, there should be some additional factor in the inequality?

Suggestions:
1. Recently, Mishchenko et al. prepared "Asynchronous SGD Beats Minibatch SGD Under Arbitrary Delays" paper. Can the authors kindly add some sentences where they describe the differences between the papers?


**Strengths And Weaknesses:**

The paper provides very interesting and nontrivial results in a timely topic. Even the fact that the authors improved dependence from $O(\frac{\sigma^2}{\epsilon^2} + \frac{\tau_{max}}{\epsilon})$ to $O(\frac{\sigma^2}{\epsilon^2} + \frac{\sqrt{\tau_{max} \tau_{avg}}}{\epsilon})$ is a strong result for a NeurIPS publication. Besides that, the authors improved the convergence rates even further to $O(\frac{\sigma^2}{\epsilon^2} + \frac{\tau_{avg}}{\epsilon})$ using adaptive step sizes. The same order of improvements was done in the heterogeneous setting.

I see that the dependence $O(\frac{\sigma^2}{\epsilon^2} + \frac{\tau_{avg}}{\epsilon})$ was obtained before by Cohen et al. "Asynchronous Stochastic Optimization Robust to Arbitrary Delays", but they use an alternative strategy with another hyperparameter that can be difficult to tune.

The proofs sound to me and I didn't find any major issues. But, I only checked the proof of Theorem 6 in detail.

---

> ### Author Response · Authors · 2022-08-02
> **Response to Reviewer 88dr**
>
> Thank you very much for your positive review !
>
> Answer on suggestion:
> 1. Thank you very much for pointing to this simultaneous work (note that it was published on ArXiv after the NeurIPS submission deadline). Mishenko et al [1] analyze a similar delay-adaptive stepsize scheme and prove a convergence rate of $O(\sigma^2 \epsilon^{-2} + n \epsilon^{-1})$, where $n$ is the number of workers, that is same as ours as $n = O(\tau_{avg})$. Similar to us, they considered asynchronous SGD with constant stepsizes, but under different assumptions: assuming only Lipschitz-continuity of functions instead of Lipschitz-smoothness. They did not discuss the connection of the number of workers to the average delay. For the heterogeneous case they chose a different approach than ours and provide a delay-adaptive learning rate that converges only to an approximate solution, but allows workers to be arbitrarily long delayed (including the case when some of the workers are never responding).
> We will cite this work and will add a discussion on the similarities and differences to the final version of our paper.
>
> Answer on minor problems:
>
> 1. We used the notation $\xi_t$ for the noise instead of $\xi_{t - \tau_t}$ to highlight that the previous iterations are independent from this noise. We agree that this might be confusing and will change notations according to your suggestion.
>
> 2. Thank you for pointing to this. We note that if $|C_T| \geq 2$, it would always hold that $\tau_{avg} \leq \tau_C$ without additional factors (with $|C_t| = 1 \forall t$ we have synchronous one-node straining); Otherwise, it always holds that $\tau_{avg} \leq 2 \tau_C$, which would degrade our convergence rate only by a constant. We will clarify this in the paper.
>
> [1] Mishenko et al, "Asynchronous SGD Beats Minibatch SGD Under Arbitrary Delays"

---

### Official Review · Reviewer_Ukik · 2022-07-10

**Rating:** 7
**Confidence:** 4
**Soundness:** 3 good
**Presentation:** 3 good
**Contribution:** 3 good

**Summary:**

This paper study the convergence properties of asynchronous SGD for distributed and federated learning. The authors tighten the convergence rate of non-convex smooth objectives that depend on the maximum delay to $\epsilon$-stationary point from $O\left(\sigma^2\epsilon^{-2}+ \tau_{\max}\epsilon^{-1}\right)$ to $ O\left(\sigma^2\epsilon^{-2}+ \sqrt{\tau_{\max}\tau_{avg}}\epsilon^{-1}\right)$ without any change to the algorithm., where $\tau_{\max}$ is maximal delay and $\tau_{avg}\$ is the average delay. Then, the authors propose a new delay-adaptive learning rate scheme of a faster convergence rate of $ O\!\left(\sigma^2\epsilon^{-2}+ \tau_{avg}\epsilon^{-1}\right)$. This is the first time that asynchronous SGD becomes faster than the synchronous SGD.  The authors also discuss the case of of heterogeneous functions.

**Questions:**

The authors are encouraged to conduct experiments to verify the advantages of the proposed delay-adaptive learning rate scheme

**Limitations:**

No limitations and potential negative societal impact are identified, while the authors do not explicitly discuss this in the paper.

**Strengths And Weaknesses:**

Pros:

+ This paper makes novel and significant contributions in asynchronous SGD for distributed and federated learning. The authors significantly reduce the reliance of the convergence on the maximal delay to the average delay, by leveraging the notion of concurrency. This decreases the worst-case impact of the communications between the workers.
+ The idea makes sense. The authors suggest that the average delay matters in determining the convergence rather than totally relying on the maximal delay.
+ The proofs are rigorous and presented clearly. I did not find major flaws.
+ The literature review is comprehensive and well-discussed. The contributions are well-positioned in the existing results. Readers can easily understand the novelty and contributions of this work compared with the existing works.
+ This paper is well-written. The paper is organised in a clear structure.

Con:

- My concern is mainly about the proposed algorithm. The authors argue that this algorithm has advanced convergence rate but does not give any experimental results. The authors are encouraged to conduct experiments to verify this statement.

---

> ### Author Response · Authors · 2022-08-02
> **Response to Reviewer Ukik**
>
> Thank you for your positive review !
>
>
> We would like to note that we already provided some experimental verification of the $\sqrt{\tau_{\max}}$ term with the constant stepsize in the appendix, section B. For the rebuttal, we updated the supplementary material (see last page) with extra comparison of delay-adaptive learning rate schemes (we will further polish this new part for the final version).

---

> > ### Comment · Reviewer_Ukik · 2022-08-07
> > **Concerns addressed**
> >
> > Thanks for your response. My concerns are addressed.

---

### Official Review · Reviewer_ZeX4 · 2022-07-11

**Rating:** 7
**Confidence:** 4
**Soundness:** 4 excellent
**Presentation:** 4 excellent
**Contribution:** 3 good

**Summary:**

This work provides a theoretic study on the convergence rates of asynchronous SGD under distributed homogeneous and heterogeneous settings. The main contributions are (i) a sharper analysis for the constant stepsize async-SGD, which improves over the existing ones, and (ii) a delay-adaptive stepsize for async-SGD, which leads to the best known convergence rate, and without extra hyperparameter tuning and extra communications required in [13], (iii) improved analysis in the heterogeneous case compared with FedBuff [31].

**Questions:**

Questions: see comments above.

Typos:
- Algorithm 1, Line 1: sever -> server.
- Line 164, C_0 =[n]?
- Line 287, there exists

**Limitations:**

The authors have adequately addressed the limitations and potential negative societal impact of their work.

**Strengths And Weaknesses:**

======= After Rebuttal =========
Thank the authors for the response, which clarified my confusions.

============================
Strengths:
- This paper is very well-written and has a nice flow of ideas.
- The key analytic technique, which I believe is the notion of active set/concurrency, seems pretty novel to me.
- Simple and intuitive algorithmic constructions with improved convergence guarantees.

Weaknesses (or comments):
- This work would benefits from some empirical evaluations, given that there is nothing impractical or hard-to-tune in the proposed schemes.
- The authors claimed that they show "for the first time, asynchronous SGD is always faster than mini-batch SGD". Ain't the convergence results in [13] (Picky SGD) already show this, did I miss something?
- Algorithm 2 does not consider local steps, which is however a key feature of federated learning systems. I am not sure if it is a straightforward extension in theory, especially with delayed updates. Could the authors discuss how the number of local steps would affect the convergence rate in Theorem 10?

---

> ### Author Response · Authors · 2022-08-02
> **Response to Reviewer ZeX4**
>
> Thank you for your positive review !
>
> 1. We would like to note that we already provided experimental verification of the $\sqrt{\tau_{\max}}$ term in the appendix, section B. For the rebuttal, we updated the supplementary material (see last page) with extra comparison of delay-adaptive learning rate schemes (we will further polish this new part for the final version).
>
> 2. The picky SGD paper [13] considered only the homogeneous setting and provided convergence rates depending on the average delay. A further difference is that they considered arbitrary delay patterns, while we focus on more realistic (or implementable) delay patterns through the notion of “concurrency” which allows us to derive tighter bounds. Without specifying concurrency n, the average delay can be larger than the number of workers n, e.g. when all the gradients are delayed by the maximum delay > n. We also note that the convergence result in [13] holds with probability ½, while we use a different proof technique to prove convergence results in expectation.
>
> 3. Thank you for your question. Local steps would worsen the convergence rates. In order to add the local steps, we need to re-define virtual sequences in the spirit of the error-feedback proofs from [47]. Lemma 19 would not change and we need to start derivations with Lemmas 20/21 that estimate the differences between “real” and “virtual” sequences. Local steps would not affect the leading convergence term. However, precise dependance on the number of local steps needs to be worked out.
>
> 4. Thank you for noting these typos !

---

### Author Response · Authors · 2022-08-02
**To all the Reviewers**

We would like to thank all of the reviewers for their very positive evaluation of our paper and their valuable comments that help to improve the paper. We provided responses to each reviewer separately below.

---

### Meta-Review · Area_Chair_vTqM · 2022-08-25

**Recommendation:** Accept
**Confidence:** Certain

**Metareview:**

The paper studies the convergence of asynchronous SGD in the setting of distributed/federated nonconvex smooth optimization with . The first contribution of the paper is to tighten the existing convergence guarantees, improving the dependence on the delay parameter from maximum ($\tau_{\max}$) to the geometric mean of maximum and average delay ($\sqrt{\tau_{\max}\tau_{\rm avg}}$). The paper then proceeds to introduce an asynchronous SGD variant with delay-adaptive learning rate that further reduces the dependence to simply $\tau_{\rm avg}.$ On a conceptual level, the paper shows that asynchronous SGD is always faster than minibatch SGD. The main techniques based on active set/concurrency seem novel and are presented in a clear way. The paper is well written, has solid contributions, and is a good fit for NeurIPS.

**Award:**

No

---

### Decision · Program_Chairs · 2022-09-14

Accept